



# Identification of volatile organic compounds emitted by Sitka spruce and determination of their emission pathways and fluxes

Hayley Furnell[1, 2], John Wenger[1, 2], Astrid Wingler[2, 3], Kieran N. Kilcawley[4, 5], David T. Mannion[4, 5], Iwona Skibinska[4, 5], Julien Kammer[1, 2, 6]

[1]School of Chemistry, University College Cork, Cork, T12 YN60, Ireland
[2]Environmental Research Institute, University College Cork, Cork, T23 XE10, Ireland
[3]School of Biological, Earth and Environmental Sciences, University College Cork, Cork, T23 TK30, Ireland
[4]Department of Food Quality and Sensory Science, Teagasc Food Research Centre, Moorepark, Fermoy, Cork, P61 C996, Ireland
[5]School of Food and Nutritional Science, University College Cork, T12 R220, Ireland
[6]Aix Marseille Université, CRNS, LCE, Marseille, 13331, France

*Correspondence to*: Julien Kammer (julien.Kammer@univ-amu.fr)

**Abstract.** The biogenic volatile organic compounds (BVOCs) emitted by Sitka spruce (Picea sitchensis) trees, housed in a plant growth chamber, were characterised by a combination of on-line (time-of-flight chemical ionisation mass spectrometry) and off-line (gas chromatography-mass spectrometry) techniques. In total, 74 BVOCs were identified in the Sitka spruce emissions, 52 of which were oxygenated compounds, with piperitone ($C_{10}H_{16}O$), an oxygenated monoterpene, being the dominant emission. Other prevalent emissions included isoprene and five monoterpenes (myrcene, β-phellandrene, δ-limonene, α-pinene, and camphene). Temperature, photosynthetic photon flux density (PPFD) and stress were all found to alter the emission profiles, with different BVOCs exhibiting different responses. Three different plant growth cycles were used to identify the emission pathways (pooled or biosynthetic) for each BVOC, through determination of the relationships of the emission flux with temperature and with PPFD. The majority of the BVOCs emitted by Sitka spruce were found to originate from biosynthetic and pooled pathways simultaneously, with those from a stressed tree having a much lower contribution from the biosynthetic pathway than a healthy tree. Standardised emission fluxes (temperature 30 °C and PPFD 1000 μmol m$^{-2}$ s$^{-1}$) were calculated for all BVOCs using the appropriate standardisation model (pooled, biosynthetic or combined). Annual emission fluxes for all Stika spruce plantations in Ireland were determined for piperitone (8,200 tonne year$^{-1}$), isoprene (13,000 tonne year$^{-1}$) and monoterpenes (1,600 tonne year$^{-1}$). At the current conditions of the Irish climate the annual BVOC flux for isoprene was found to exceed that for piperitone, although this is expected to change in a warming climate.

## 1 Introduction

Sitka spruce (*Picea sitchensis*) is native to the west coast of North America (Geron et al., 2000). Owing to its rapid growth in maritime climates with moist soils, it is commonly used in forestry plantations in north-western Europe as part of afforestation programmes. For example Sitka spruce accounts for over 50% of forestry by area in Ireland (Teagasc, 2023) and over 26% in





the UK (Forest Research, 2023). Forestry plantations have environmental impacts on the local area, influencing both biodiversity (O'Callaghan et al., 2017; Williams et al., 2017) and climate (Carslaw et al., 2010; Mahilang et al., 2021). Each different plant species emits a unique mixture of biogenic volatile organic compounds (BVOCs) that undergo multiple

atmospheric oxidation reactions to produce secondary organic aerosols (SOA), which contribute to the regional and global climate forcing (Hallquist et al., 2009; Shrivastava et al., 2017). Therefore, it is important to determine the BVOC emission profile of plant species prior to establishing large scale plantations.

BVOCs can be emitted by plants through a biosynthetic pathway, pooled emission pathway or a combination of both

mechanisms. In the biosynthetic pathway, BVOCs are synthesised within the chloroplasts of the plant and are emitted immediately (Guenther et al., 1991). Biosynthetic pathways only operate under illumination, and are strongly influenced by photosynthetically active radiation (PAR) (Tingey et al., 1979), measured as photosynthetic photon flux density (PPFD), as well as temperature (Guenther et al., 1993). In the pooled emission process, BVOCs are synthesised, deposited in storage pools and released later (Guenther et al., 1993). The emission of BVOCs from storage pools is not dependent on PAR and occurs

under both light and dark conditions. BVOC emissions from pooled sources have a strong relationship with temperature (Kesselmeier and Staudt, 1999). The BVOCs are assumed to diffuse out of the storage pools, with the rate of diffusion related to the vapour pressure of the particular BVOC (Schuh et al., 1997). PPFD and temperature are the main drivers for BVOC emissions, although other environmental conditions and parameters also contribute, including precipitation, season, age and stress (Peñuelas et al., 1999; Hakola et al., 2001; Mofikoya et al., 2018).


Numerous field and laboratory studies have been conducted to determine the composition of BVOC emissions, and to probe the factors (temperature, PPFD, stress etc.) that influence their release into the atmosphere. Although not conducted under real-world conditions, laboratory studies have the advantage of providing a controlled environment for investigating the parameters that affect BVOC emissions in a systematic manner (Street et al., 1993a).


BVOC emission studies are typically conducted over a range of environmental conditions which leads to significant variation between emission fluxes, especially when measurements are taken at different temperature and PPFD conditions. To facilitate comparison between studies, BVOC emission fluxes are reported at standard conditions of 30 °C and 1000 μmol m$^{-2}$ s$^{-1}$. For studies conducted under different temperature and PPFD conditions a standardisation procedure is applied to extrapolate the

collected data to standard conditions (Guenther et al., 1993).

To date, a limited number of studies have been conducted on the BVOC emissions from Stika spruce. The monoterpenes myrcene and β-phellandrene were found to dominate the BVOC emissions (Geron et al., 2000), with a significant contribution also from isoprene (Street et al., 1996). The earlier studies focussed primarily on the detection of hydrocarbons using offline

GC/MS techniques, with only the most recent study using proton transfer reaction-mass spectrometry (Hayward et al., 2004)





for online measurements. Recent advances in the instrumentation used for detecting ambient VOCs has led to the development of the high resolution online time-of-flight chemical ionisation mass spectrometer (ToF-CIMS) (Bertram et al., 2011), which allows for the real-time detection of a wide range of hydrocarbon and oxygenated VOCs (Kim et al., 2016; Riva et al., 2019), that in the context of Sitka spruce emissions would not have been detected in previous studies.


Given the dominance of Sitka spruce within afforestation programmes, further work is required to identify the BVOC emissions from Sitka spruce and to assess their climatic impacts. The aims of this work were (i) to use a ToF-CIMS and thermal desorption gas chromatography mass spectrometer for identification of the BVOCs emitted from Sitka spruce, in a laboratory setting under different environmental conditions, (ii) to determine the emission pathways and standardised emission fluxes for each BVOC, and (iii) to calculate annual emission fluxes for the main BVOCs.

## Materials and Methods

### 2.1 Experimental set-up

Three Sitka spruce trees were included in this study. The trees were grown from seed since spring 2017, at Fermoy Woodland Nursery, Co. Cork. In January 2020, they were placed in cold storage, where they took on a state of dormancy. In July 2020, the trees were planted in individual 5 L pots containing a peat soil, and immediately placed inside a plant growth chamber (Panasonic MLR-352). The trees were removed from the plant growth chamber in November 2020 for two months and placed in front of a south-facing window, before being reinstalled in January 2021 for emissions sampling. For identification purposed the trees were named; Spruce 1, Spruce 2 and Spruce 3 (Fig. S1).


The overall set-up for emissions sampling is shown schematically in Fig. 1 and is based on dynamic emissions chambers described previously (Niinemets et al., 2011). Filtered outdoor air was passed over the trees housed in individual Teflon enclosures inside the plant growth chamber. The two interior side walls and door of the plant growth chamber were each equipped with five photosynthetically activating lamps. The lamps extended from the floor to the ceiling and provided a PPFD range from 0 $\mu mol^{-2}$ $s^{-1}$ (no lamps illuminated) and 250 $\mu mol^{-2}$ $s^{-1}$ (all lamps illuminated when empty). The temperature was adjustable between 5 °C and 40 °C. Further details of the set-up are provided in the Supplement.



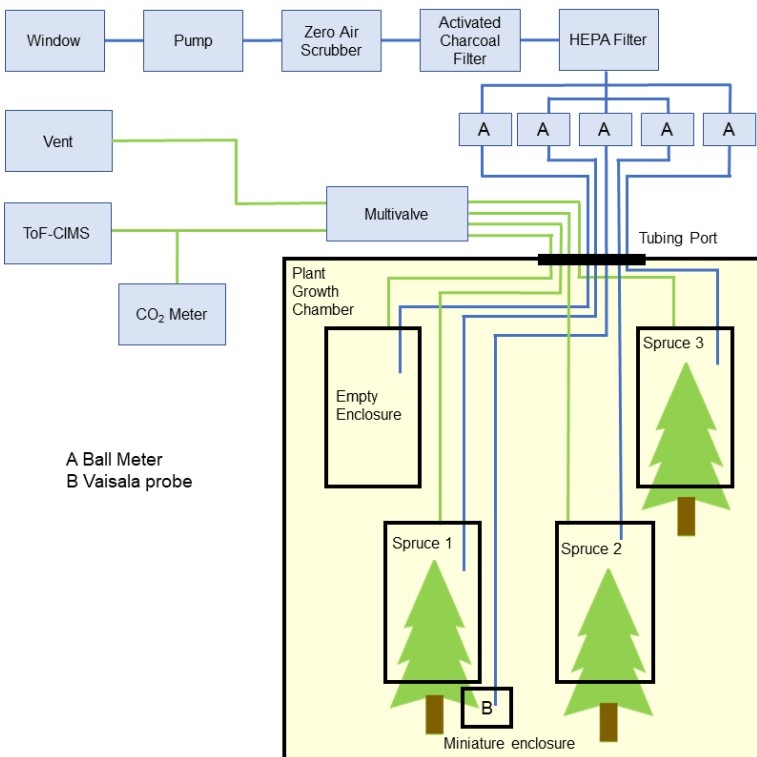

**Figure 1: Schematic of emissions sampling set-up.**


Outdoor air was sampled through an open window at 32 L min$^{-1}$ with a pump (Becker VT4.4) and passed though a zero air scrubber (OPT 86C, Teledyne) and a home-made activated charcoal filter (Hatchwells granulated charcoal) to remove atmospheric oxidants and VOCs, as well as a HEPA filter (TSI) to remove particles. The filtered air flow into each enclosure was set to a fixed value between 1.5–7 L min$^{-1}$, depending on the degree to which the enclosure was sealed.


On several occasions, very high concentrations of an unknown interferant (ToF-CIMS ion $C_{10}H_{15}^{+}$) were detected in the outdoor air, which distorted the BVOC measurements. During periods of interference the sample line was temporarily moved indoors and allowed to sample humidified compressed air until the interference passed.

Prior to reinstallation in the plant growth chamber in January 2021, all trees had dark green needles, with Spruce 1 having the largest amount of foliage and Spruce 3 the least. Spruce 1 and Spruce 3 had visible resin deposits on their tree trunks (Fig. S2). A month prior to emissions sampling Spruce 1 produced new shoots, the majority of which were outside the enclosure. The trees were watered twice a week with 350 mL of tap water each and did not receive any fertiliser treatment.



Temperature and relative humidity measurements (dewpoint probe DMP74A, Viasala) were taken from inside a miniature

enclosure to account for any changes in conditions due to the presence of the enclosure (Gomez et al., 2021). The miniature

enclosure was sealed around one of the lower branches of Spruce 1 and was supplied with the same filtered air.

The use of a flow-through multivalve (Multiposition Actuator EMTMA-CE, Vici Valco) allowed rapid and controlled sample

switching between the various enclosures. The outflows from the three spruce enclosures and empty enclosure were each

connected to an inlet on the multivalve. At any time a single inlet was connected to the sample port which supplied the ToF-

CIMS and $CO_2$ meter, while the remaining inlets were connected to the exhaust which was externally pumped to ensure that

all lines were continually flushed to prevent dead volume effects. The multivalve was programmed to connect the sample port

with each inlet for 7.5 min, in the following order: empty enclosure, Spruce 1, Spruce 2, Spruce 3. This 30 min cycle was

continuously repeated. The remaining flows were vented to avoid dead volume and tubing memory effects.

## 2.2 Instrumentation

### 2.2.1 ToF-CIMS

A high resolution time-of-flight chemical ionisation mass spectrometer (ToF-CIMS, Aerodyne C4Q-106) was used to monitor

the BVOC emissions from the Sitka spruce trees in real time. The basic operation of the ToF-CIMS has been described

elsewhere (Bertram et al., 2011; Lee et al., 2014). The ToF-CIMS used in this work was fitted with a VUV ionisation source

(Ji et al., 2020). The benzene cation ($C_6H_6^+$) was selected as the reagent ion for these measurements because of its ability to

ionise compounds with a low oxygen content, as well as hydrocarbons such as monoterpenes, which are typical of biogenic

emissions (Lavi et al., 2018). Benzene vapour was generated by flowing $N_2$ (BOC, oxygen free) at a rate of 0.25 L min$^{-1}$ over

a liquid benzene permeation tube (6-PD-1400-C45, Carl Stuart Group) placed inside ¾" stainless steel tubing, which was

maintained at 50 °C. The vapourised benzene passed into the VUV source where upon absorption of UV radiation, $C_6H_6^+$ was

formed via the ejection of an electron (Breitenlechner et al., 2022). The voltages and pressures applied to the ToF-CIMS were

controlled via TOFWERK TPS Settings. Trichlorobenzene (Sigma Aldrich, 99%), was used as an internal mass calibrant, by

allowing it to diffuse into the inlet of the instrument.


To quantify BVOCs, calibration coefficients were determined for δ-3-carene, β-myrcene, camphor (Sigma Aldrich, purity

90%, 75% and 98%, respectively) and piperitone (Santa Cruz Biotechnology, purity > 94%), by injecting aliquots of BVOC

into a 2.2 m$^3$ atmospheric simulation chamber at the Environmental Research Institute, University College Cork (Ashu-Ayem

et al., 2012). The concentrations of BVOCs from the Sitka spruce trees were below the range of the linear calibration plots,

and it was therefore assumed that the ToF-CIMS continued to have a linear response over the full range of concentrations



measured during BVOC emissions sampling and calibrations (Lee et al., 2014). The calibration coefficients for the four BVOCs were applied to various ions detected with the ToF-CIMS according to their physiochemical properties.

Data collected by the ToF-CIMS were analysed with Tofware 3.2.0, operated through IGOR Pro 7.08. Firstly, 1 Hz data were averaged to a 10 s time resolution. Following this, the recommended Tofware calibration and correction steps were performed. Chemical formulae were then assigned to peaks within the calibrated mass range. Only formulae containing the elements C, H, O, N and Cl were assigned. If an appropriate formula could not be determined, the peak was left unassigned, and excluded from further analysis. After completing peak assignment, data points recorded during periods of interference (disconnection of the $CO_2$ meter and contamination from $C_{10}H_{15}^+$) were removed.

### 2.2.2 TD/GC-MS

Sitka spruce emissions were collected on Tenax Carbograph thermal desorption tubes (Markes International Ltd.) conditioned prior to sampling. Tubes capped with difflock caps were placed on the tray of an autosampler (Unity 2 Autosampler, Markes International Ltd.), subsequently introduced to the Unity 2 Thermal Desorption Unit (Markes International Ltd.), which was connected by a heated transfer line to a split/splitless injector of an Agilent 7890A GC and Agilent 5977B MDS gas chromatograph mass spectrometer (GC/MS) (Agilent Technologies). The first step of the desorption involved dry purging the tubes for 2 min using a 1:20 split with $N_2$ at 50 psi, before undergoing a 5 min thermal desorption at 150 °C, which was followed by a secondary desorption at 280 °C for 5 min. BVOCs were collected onto a materials emissions trap and this was held at 30 °C during the tube desorption with a gas flow of 50 ml min$^{-1}$. A 2 min fire purge was applied to the trap with a 1:50 split, the BVOCs were desorbed from the trap using a temperature ramp of 24 °C min$^{-1}$ with a 1:10 split for 5 min. The transfer line was heated to 160 °C, the GC injector was set at 250 °C and operated in splitless mode. The GC/MS was fitted with a 60 m capillary column (Agilent Technologies Ltd, DB 624UI 60 m × 0.3 mm × 1.8 μm). the carrier gas was helium and set at a constant pressure of 23 psi. The column temperature was initially held at 40 °C for 5 min, and then increased at a rate of 5 °C min$^{-1}$ to 230 °C, held for 35 min with a total run time of 78 min. The mass spectra were generated by quadrupole MS detector with ionisation voltage of 70 eV, 3.32 scans s$^{-1}$. The ion source temperature was 230 °C and the interface temperature was set at 280 °C.

The chromatograms generated from each tube were analysed with MassHunter Qualitative Analysis Navigator (B.08.00, 2016). BVOC identification was verified using retention indices and fragmentation patterns compared to an in-house spectral library and the NIST 2014 mass spectral library with the assistance of deconvoluting software AMDIS (v2.72, June 2014). Linear Retention indices were determined using established methods (Van Den Dool and Dec. Kratz, 1963) In addition, BVOC isomers in the Sitka spruce emissions were separated by TD-GC/MS and identified with the use of standards. The presence of isoprene, α-pinene, β-pinene, myrcene, camphene, camphor (Sigma Aldrich, purities 99%, 98%, 99%, 75%, > 96% and 98%





respectively) and piperitone (Santa Cruz Biotechnology, purity 94%) were verified in the Sitka spruce emissions (Evans et al.,
1985; Tani et al., 2003). The BVOCs were dissolved in methanol and subsequently in distilled water to obtain a mixing ratio
of 10 ppm. A 10 µL volume of each standard was transferred onto Tenax tubes and analysed on the TD-GC/MS system at the
same condition as Sitka spruce samples. An auto-tune of the GC/MS was carried out prior to the analysis to ensure optimal
GC/MS performance. A set of external standards was run at the start and end of the sample set and abundances were compared
to known amounts to ensure that both separation and MS detection was performing within specification.


### 2.2.2 Ancillary Measurements

A $CO_2$ meter (K30 ELG 10,000 ppm $CO_2$ Data Logging Sensor) was used to monitor $CO_2$ uptake and emission by the trees,
to determine photosynthetic activity and respiration. The operation of the $CO_2$ meter is based on near infra-red absorption and
was controlled with GasLab software, which recorded a measurement every 10 s. The sensor background was controlled using
pure $N_2$ (BOC, oxygen free) and a calibration up to 911 ppm was performed by adding pure $CO_2$ (BOC, 99%) into an
atmospheric simulation chamber.

A light meter (LI-250, LI-COR Light Meter) was used to measure PPFD in the centre of the plant growth chamber with all
three Sitka spruce trees and enclosures present (Table S4). The measurements were taken before and after the emissions
sampling cycles. The plant growth chamber door remained closed during PPFD measurements. The average PPFDs recorded
at the three light settings were 21 µmol m$^{-2}$ s$^{-1}$, 44 µmol m$^{-2}$ s$^{-1}$ and 164 µmol m$^{-2}$ s$^{-1}$.

A chlorophyll fluorescence meter (Hansatech FMS2) was used to determine maximum photosystem II efficiency ($F_v/F_m$) of
the Sitka spruce. Chlorophyll fluorescence measurements were recorded twice a day, three days per sampling cycle. One
measurement was taken 30 min prior to illumination and a further measurement was taken 90 min into the maximum PPFD
and temperature setting. Three branches on each tree were selected for measurements. Clips were placed over the needles for
dark adaptation for 30 min before a measurement.

### 2.3 Experimental Procedures

Three different 24-hour plant growth cycles were used to assess the impact of temperature and PPFD on the BVOC emissions
from the Sitka spruce trees. Prior to BVOC emission sampling an acclimation period of 2–3 days was applied to allow the
trees sufficient time to adapt to the new conditions. As the emissions were sampled from only three Sitka spruce trees,
measurements were recorded for a minimum of 7 consecutive days to maximise reproducibility for statistical purposes.



For the *Daily Cycle* the trees were subject to conditions as similar as possible to Irish summer conditions, in which both temperature and PPFD increase simultaneously to a maximum and decrease. The purpose of the *Daily Cycle* was to determine the BVOC and $CO_2$ emissions from the Sitka spruce trees, and to assess variations in BVOC emissions and photosynthesis in response to changes in PPFD and temperature. The conditions for the *Daily Cycle* were a repeating 24 hour cycle of: 10 hours of darkness at 12 °C; 3 hours at 21 μmol m$^{-2}$ s$^{-1}$ and 15 °C; 3 hours at 44 μmol m$^{-2}$ s$^{-1}$ and 18 °C; 2 hours at 164 μmol m$^{-2}$ s$^{-1}$

and 21 °C; 3 hours at 44 μmol m$^{-2}$ s$^{-1}$ and 18 °C; and 3 hours at 21 μmol m$^{-2}$ s$^{-1}$ and 15 °C (Fig. S4).

The purpose of the *Temperature Cycle* was to determine the effect of temperature alone on BVOC emissions and photosynthesis. During the hours of illumination a constant PPFD of 44 μmol m$^{-2}$ s$^{-1}$ was applied (Schuh et al., 1997). The conditions for the *Temperature Cycle* were: 4 hours of darkness at 12 °C; 3 hours of darkness at 15 °C; 3 hours of illumination

at 15 °C; 3 hours of illumination at 18 °C; 2 hours of illumination at 21 °C; 3 hours of illumination at 18 °C; 3 hours of illumination at 15 °C; and 3 hours of darkness at 15 °C (Fig. S5).

The purpose of the *Light Cycle* was to identify changes in BVOC emissions and photosynthesis resulting from variations in PPFD only. A constant temperature of 18 °C was applied for the duration of the *Light Cycle*. During the *Light Cycle* PPFD

was varied according to: 10 hours of darkness; 3 hours at 21 μmol m$^{-2}$ s$^{-1}$; 3 hours at 44 μmol m$^{-2}$ s$^{-1}$; 2 hours at 164 μmol m$^{-2}$ s$^{-1}$; 3 hours at 44 μmol m$^{-2}$ s$^{-1}$; and 3 hours at 21 μmol m$^{-2}$ s$^{-1}$ (Fig. S6).

After completion of the three plant growth cycles, two samples from each enclosure were collected onto Tenax tubes (Markes Tenax TA, C1-CAXX-5003) at a flow rate of 200 mL min$^{-1}$ for 2 hours. Four samples (one sample from each enclosure) were

collected with conditions in the plant growth camber set to 44 μmol m$^{-2}$ s$^{-1}$ and 18 °C and the other four samples were collected with conditions in the plant growth chamber set to 164 μmol m$^{-2}$ s$^{-1}$ and 24 °C. To account for any potential residue remaining on the Tenax tubes from previous use, two tubes were left unopened and deemed to be sample blanks. Sampling onto the Tenax tubes did not commence until an hour after the conditions in the plant growth chamber had been set (Street et al., 1996). Following collection the Tenax tubes were sealed, individually wrapped in aluminium foil, placed into separate zip lock bags,

and stored in a refrigerator (Staudt et al., 2003; Hellén et al., 2023) until analysis could be performed.

Following all measurements, the Sitka spruce trees were cut, and all biomass contained within the enclosures was removed, and dried at 60 °C for 72 hours (Hayward et al., 2004) using incubators (Memmert IN110 Incubator). When drying was completed a mass balance (Explorer OHAUS, E02140) was used to determine the dry mass of the needles only, as well as the

dry mass of the needles plus branches for each tree (Table S1).



## 2.4 Emission calculation and modelling

The BVOC and $CO_2$ fluxes for each measurement cycle were analysed using R Studio 4.0.2. The last 2 min of the 7.5 min sampling period for each of the four enclosures were used to generate 30 min averages. The $CO_2$ fluxes were calculated for
each spruce tree according to Eq. (1):

$$Flux_{CO_2} = \frac{(Spruce_{CO_2} - empty_{CO_2}) \times Factor_{CO_2}}{N_A \times Biomass}, \tag{1}$$

where $Flux_{CO_2}$ is the $CO_2$ flux in nmol s$^{-1}$ g$^{-1}$; $Spruce_{CO_2}$ is the $CO_2$ mixing ratio (in ppm) in the Spruce enclosure; $empty_{CO_2}$ is the $CO_2$ mixing ratio (in ppm) in the empty enclosure; $Flow$ is the flow through the Spruce enclosure in m$^3$ h$^{-1}$; $N_A$ is Avogadro's number in mol$^{-1}$; $Biomass$ is the mass of the needles in g$_{dw}$; and $Factor_{CO_2}$ is a constant in h ppm$^{-1}$ m$^{-3}$ s$^{-1}$, which
accounts for unit conversions.

A student's t-test was performed on the data from the *Daily Cycle* to differentiate ions due to the BVOCs from those in the background spectra. Only ions with a p-value of less than 0.05 when compared with three times the ion signal from the empty enclosure, measured at a PPFD greater than 40 μmol m$^{-2}$ s$^{-1}$, were attributed as statistically significant emissions. Only the ions
identified in the *Daily Cycle* were analysed for the *Temperature Cycle* and *Light Cycle*. No ion signals from Spruce 3 were above the statistical threshold, and therefore Spruce 3 was removed from all further analysis.

The concentrations of the BVOCs emitted by each spruce tree were then converted to emission fluxes using Eq. (2):

$$Emission\ Flux_{BVOC} = \frac{(Spruce_{BVOC} - Empty_{BVOC}) \times Flow \times Mw_{BVOC} \times Factor}{N_A \times Biomass}, \tag{2}$$

where *Emission Flux*$_{BVOC}$ is the flux of a specific BVOC from the spruce tree in μg g$_{dw}^{-1}$ h$^{-1}$; $Spruce_{BVOC}$ is the concentration of the BVOC from the spruce tree in ions s$^{-1}$; $Empty_{BVOC}$ is the concentration of the BVOC from the empty enclosure in ions s$^{-1}$; $Flow$ is the flow through the Sitka spruce enclosure in m$^3$ h$^{-1}$; $Mw_{BVOC}$ is the molecular weight of the BVOC in g mol$^{-1}$; $Biomass$ is the mass of the dried needles and branches in g$_{dw}$; and $Factor$ is a constant to account for unit conversions (Hayward et al., 2004) in ions$^{-1}$ s$^{-1}$ m$^{-3}$.


Spruce BVOC emissions have been reported to originate through pooled sources (monoterpenes) and biosynthetic pathways (isoprene) (Hayward et al., 2004; Street et al., 1996). The emission fluxes were standardised to a PPFD of 1000 μmol m$^{-2}$ s$^{-1}$, and a temperature of 30 °C according to Eq. (3), (6) and (7) developed by Guenther et al., (1993) and adapted by Schuh et al., (1997):

$E_{isoprene} = E_{isoprene}^{standard} \times C_L \times C_T, \tag{3}$



where $E_{\text{isoprene}}$ is the isoprene emission flux at the measured temperature and PPFD in µg $g_{dw}^{-1}$ $h^{-1}$; $E_{\text{isoprene}}^{\text{standard}}$ is the isoprene emission flux in µg $g_{dw}^{-1}$ $h^{-1}$ at standard conditions of 30 °C and 1000 µmol $m^{-2}$ $s^{-1}$; $C_L$ and $C_T$ are factors to account for the PPFD and temperature dependence of the emissions respectively. $C_L$ is described by Eq. (4):

$$C_L = \frac{\alpha \times c_{L1} \times L}{\sqrt{1 + (\alpha^2 \times L^2)}}, \tag{4}$$

where $\alpha$ and $c_{L1}$ are empirical constants with values 0.0027 and 1.066 respectively; and $L$ is the PPFD in µmol $m^{-2}$ $s^{-1}$. The temperature dependence of biosynthetic emissions is described by Eq. (5):

$$C_T = \frac{\exp\left(\frac{c_{T2} \times (T - T_S)}{R \times T \times T_S}\right)}{1 + \exp\left(\frac{c_{T3} \times (T - T_M)}{R \times T \times T_S}\right)}, \tag{5}$$

where $c_{T2}$ is an empirical constant of value 95,000 J $mol^{-1}$; $T$ is the measured temperature in K; $T_S$ is standard temperature, 303 K; $R$ is the universal gas constant, 8.314 J $K^{-1}$ $mol^{-1}$; $c_{T3}$ is an empirical constant of value 230,000 J $mol^{-1}$; and $T_M$ is the
temperature of maximum enzyme activity in K. This equation has been used in multiple studies to standardise isoprene emissions (Gomez et al., 2021; Janson et al., 1999).

The emission of monoterpenes is reported to be solely temperature dependent (Street et al., 1996), and is described by Eq. (6):

$$M = M_s \times \exp(\beta \times (T - T_s)), \tag{6}$$

Where $M$ is the measured monoterpene emission flux in µg $g_{dw}^{-1}$ $h^{-1}$; $M_s$ is the monoterpene emission flux in µg $g_{dw}^{-1}$ $h^{-1}$ at standard temperature, $\beta$ is an empirical coefficient of value 0.09 $K^{-1}$, $T$ is the measured temperature in K; and $T_s$ is standard temperature, 303 K.

Studies by Schuh et al. (1997) on sunflower and beech found BVOC emissions to originate from a combined biosynthetic and
pooled emission pathway. A combined equation was developed under the assumption that both emission pathways are completely independent, and the total emission flux is the sum of the emission flux from each pathway, described by Eq. (7):

$$E_{combined} = E_{pooled} + E_{biosynthesis}, \tag{7}$$

where $E_{\text{combined}}$ is the total emission flux in µg $g_{dw}^{-1}$ $h^{-1}$, $E_{\text{pooled}}$ is the temperature only dependent emission flux from storage pools in µg $g_{dw}^{-1}$ $h^{-1}$, using the monoterpene emission equation $M$ developed by Guenther et al. (1993): and $E_{\text{biosynthesis}}$ is the
biosynthetic emission flux, using the isoprene emission equation $E_{\text{isoprene}}$ developed by Guenther et al. (1993). The coefficients in each equation were derived using the measurement data from sunflower and beech.




# 3 Results and discussion

## 3.1 Identification of emitted BVOCs

In total 74 BVOCs were detected in the emissions from all three Sitka spruce trees using the ToF-CIMS and TD-GC/MS. To date, a relatively small number of BVOCs have been reported in the emissions from Sitka spruce. However, in this work 49 different BVOCs were detected in the emissions from Spruce 1 and 58 in the emissions from Spruce 2 (Fig 2 and S2). A month before sampling commenced Spruce 1 produced new shoots. The new shoots were a bright green colour, while the needles were soft, and mainly on the lower part of the trunk outside the enclosure. At the time of sampling the needles had darkened

to almost the same dark green colour as the older needles. Spruce 2 had less foliage than Spruce 1, and all needles were dark green. While there was some variation between the trees, there was also a degree of consistency, with each tree emitting the same 21 BVOCs.

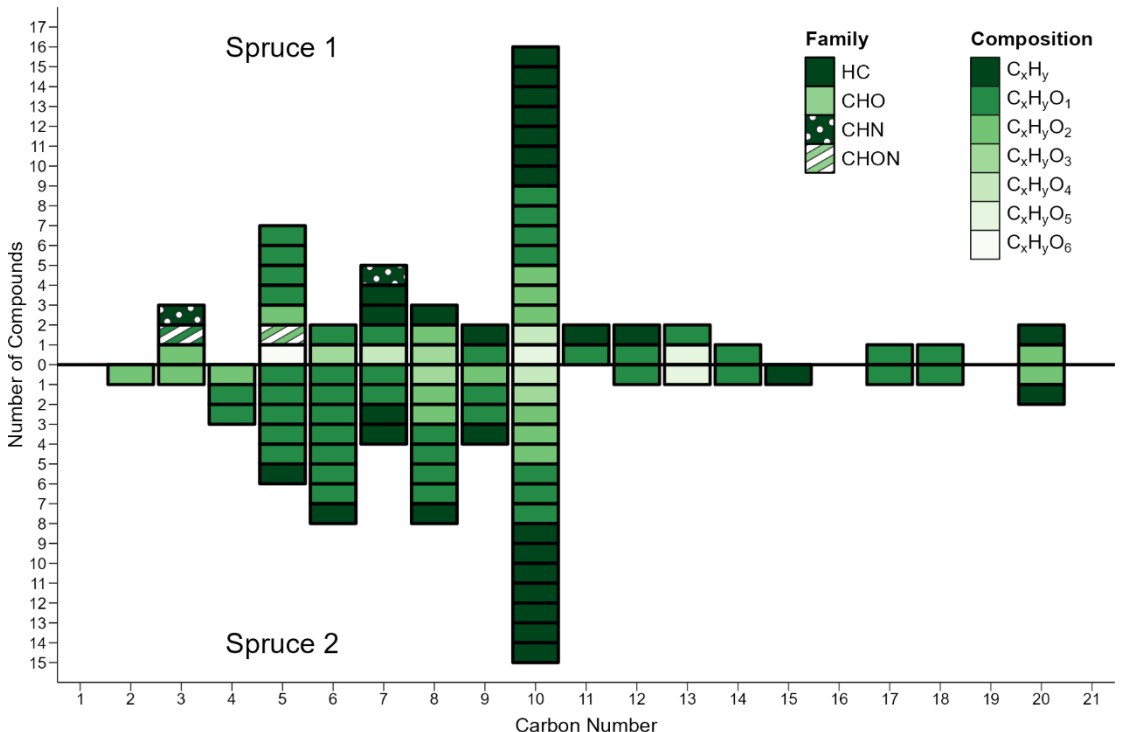

**Figure 2: BVOCs emitted by Spruce 1 (top) and Spruce 2 (bottom).**

The majority, 70%, of the BVOCs detected from all three spruce trees were oxygenated compounds. This observation is somewhat different from reports in the literature, which typically identify isoprene (Purser et al., 2021), monoterpenes (Janson, 1993) and sesquiterpenes (Haapanala et al., 2009) as the dominant biogenic emissions. The contribution from oxygenated





BVOCs is usually quite low, and reporting is typically restricted to specific groups: small compounds such as acetone (Fuentes et al., 2000) and methanol (Seco et al., 2007); green leaf volatiles (GLVs) (Scala et al., 2013); and the $C_{10}H_{18}O$ isomers, including linalool (Helmig et al., 2006) and eucalyptol (Hakola et al., 2006). Few studies have reported oxygenated BVOCs that fall outside these groups (Kim et al., 2010; Praplan et al., 2020). The high proportion of oxygenated BVOCs detected in this study is most likely due to the measurement capabilities of the instrumentation used. In particular, $C_6H_6^+$ ToF-CIMS is

suited to the detection of hydrocarbons and certain oxygenated compounds (Kim et al., 2016). In contrast, GC-MS and PTR-MS which are traditionally used in the analysis of BVOCs, tend to focus on the detection of hydrocarbons and small oxygenated compounds (Vanreken et al., 2006; Street et al., 1996).

An oxygenated monoterpene, piperitone ($C_{10}H_{16}O$) was the main BVOC emitted from Spruce 1 and the second highest

emission from Spruce 2. Piperitone has been identified as a BVOC emission in previous studies, but never as the main emitted BVOC. Piperitone was identified as a minor emission from damaged Sitka spruce branches, with the sum of the concentration of the isomers piperitone and camphor found to be over an order of magnitude lower than the sum of monoterpene concentration (Tani et al., 2003). The elevated piperitone concentration in this work may be related to differences in environmental conditions between this work and other studies, or due to different chemotypes, as previously reported for Scots

pine (Ylisirnio et al., 2020). Piperitone has also been detected as a BVOC emitted by Norway spruce (Kännaste et al., 2008) and Ponderosa pine (Helmig et al., 2013).

Several studies on the needle oil (Hrutfiord et al., 1974; Von Rudloff, 1978; Brooks et al., 1987) and cortical resin (Von Rudloff, 1978) of Sitka spruce found it to be composed mainly of myrcene ($C_{10}H_{16}$), and two $C_{10}H_{16}O$ isomers, piperitone and

camphor. In all studies, the proportion of piperitone in the leaf oil and cortical resin was always greater than for camphor. The detection of camphor in the BVOC emissions of Sitka spruce (Evans et al., 1985; Tani et al., 2003), is more common than for piperitone, implying that in the emissions camphor is the dominant isomer. However, in this work, piperitone was much more abundant than camphor and the ratio of piperitone to camphor was similar to that reported in the literature for resin. Importantly, resin deposits were visible on the trunk of Spruce 1 and may also have been present on Spruce 2 (Fig. S2). The

high piperitone to camphor ratio observed in this study therefore indicates that evaporation from the resin deposits may have contributed to the BVOC emissions detected from the Sitka spruce trees.

The second highest emission from Spruce 1 was isoprene ($C_5H_8$), but for Spruce 2 it was not present in the ToF-CIMS analysis and was only detected at trace levels in the TD-GC/MS analysis. A similar phenomenon has been observed for Scots pine, in

which $\delta$-carene was a dominant emission from one chemotype, and not emitted from the other (Ylisirnio et al., 2020). This further supports the idea that trees of the same species can have different emission profiles. Isoprene has been detected in the emissions of various spruce species, including Sitka spruce (Evans et al., 1985; Hayward et al., 2002; Street et al., 1996), Norway spruce (Bourtsoukidis et al., 2014; Filella et al., 2007), and several others (Kesselmeier and Staudt, 1999; Kempf et



al., 1996). Isoprene has been reported as a minor emission for other coniferous species including Scots pine (Janson et al.,
1999), although is it more commonly found in the emissions from deciduous species such as Holm oak (Lang-Yona et al.,
2010) and willow (Hakola et al., 1998).

The sum of monoterpenes ($C_{10}H_{16}$) was the third highest emission from Spruce 1, and the highest from Spruce 2. Five
monoterpenes were detected, myrcene, β-phellandrene, α-pinene, δ-limonene and camphene, all of which have previously been
reported as Sitka spruce emissions. The same isomers were detected from both trees, although the relative contributions were
slightly different. In agreement with previous studies (Geron et al., 2000; Purser et al., 2021; Street et al., 1996), myrcene was
the dominant monoterpene emitted from both trees (inferred from chromatogram peak areas), followed by β-phellandrene
(Street et al., 1996; Geron et al., 2000), which has also been previously detected in the emissions of other spruce species (Evans
et al., 1985; Geron et al., 2000). The contributions from α-pinene (Purser et al., 2021; Evans et al., 1985; Hewitt and Street,
1992) and δ-limonene also made a substantial contribution to the monoterpene emissions from Spruce 2; in contrast to the
monoterpene mixture emitted from Spruce 1 which had higher amounts of α-pinene than δ-limonene. The contribution from
camphene (Purser et al., 2021; Evans et al., 1985; Hewitt and Street, 1992) was low from both trees. Some studies have reported
β-pinene (Street et al., 1996; Evans et al., 1985; Tani et al., 2003) as a major emission from Sitka spruce, but it was not
identified by TD-GC/MS in this work.


The third highest emission from Spruce 2, $C_6H_8O$, was not detected in the emissions of Spruce 1 and a structure was not
confirmed by TD-GC/MS. In total, six of the emissions that were unique to Spruce 2 are $C_6$ compounds, some of which are
known to be GLVs (Scala et al., 2013), such as *E*-2-hexenal (Mäntylä et al., 2008). GLVs are emitted from plants in response
to stress, such as wounding (Portillo-Estrada and Niinemets, 2018; Hellén et al., 2021), infestation (Mäntylä et al., 2008) or
drought (Mentel et al., 2013). The $C_6H_8O$ compound may also be a GLV, emitted by Sitka spruce in response to stress. In
addition, a sesquiterpene, *E*-β-farnesene, was detected in the Spruce 2 emissions. Elevated levels of sesquiterpenes have been
observed in the emissions of ozone stressed plants (Heiden et al., 1999; Bourtsoukidis et al., 2012). In other studies the emission
of farnesenes from Norway spruce (Kännaste et al., 2008) and Scots pine (Ylisirnio et al., 2020) were found to be elevated due
to infestation. Although all trees in this work were exposed to the same conditions, the presence of $C_6$ GLVs and *E*-β-farnesene
indicate that Spruce 2 may have been stressed. Low photosystem II efficiency ($F_v/F_m$) values are commonly used to indicate
chronic photoinhibition caused by plant stress. However, for Spruce 2 the $F_v/F_m$ values were similar to those for Spruce 1 and
were not indicative of Spruce 2 suffering stress (Tables S3, S4 and S5). This implies that the stress experienced by Spruce 2
was a mild stress that mainly affected the BVOC synthesis pathways, but without causing chronic photoinhibition.



## 3.2 Influence of environmental parameters

The ToF-CIMS results obtained during the *Daily Cycle* show that all BVOC emissions responded to the simultaneous changes in temperature and PPFD as expected. The BVOC emission fluxes were at their highest when temperature and PPFD peaked and were at their lowest during the hours of darkness at the lowest temperature. Fig. 3 shows the time series profile for the emission of piperitone from Spruce 1 during the *Daily Cycle*. The time series for most other BVOCs emitted from Spruce 1 followed a similar pattern.

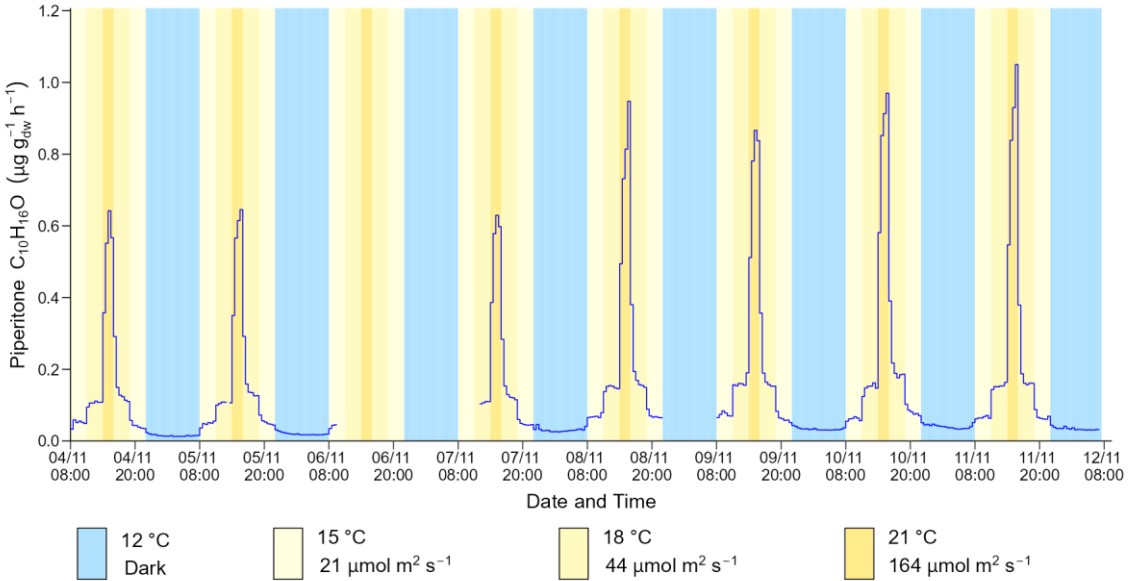

**Figure 3: Time series of the piperitone emission flux from Spruce 1 during the Daily Cycle.**

The emission of piperitone from Spruce 1 was lowest during the hours of darkness at 12 °C. Upon illumination of the plant growth chamber to 21 $\mu$mol m$^{-2}$ s$^{-1}$ and temperature increase to 15 °C, the emission of piperitone increased immediately and reached a new steady state. The emission increased with a further 3 °C rise in temperature and doubling of the PPFD. A sharp increase in emission was observed following the transition to the most intense conditions of 21 °C and 164 $\mu$mol m$^{-2}$ s$^{-1}$. The emission flux took longer to stabilise at these settings, likely due to the significant increase in PPFD. As the conditions were cycled back to darkness, the BVOC emission fluxes decreased accordingly. For a given temperature and PPFD the emission flux attained a certain value, irrespective of short-term history. The magnitude of the change in flux at the different conditions depended on the specific BVOC, with some species showing a pronounced increase in emission upon illumination, and others having a more muted response. The relationships between BVOC emission fluxes and light and temperature have been reported for many tree and plant species (Laothawornkitkul et al., 2009; Street et al., 1996). In the field, BVOC emissions have been





reported to be at their lowest at night, and highest during the day when temperature and PPFD reach a maximum (Bourtsoukidis et al., 2014), as observed in this work.

The time series profile shown in Fig. 3 is representative of the emission profile for 28 of the 34 BVOCs detected by ToF-CIMS in the Spruce 1 emissions. While the remaining six BVOCs - isoprene ($C_5H_8$), $C_5H_8O_2$, $C_{11}H_{14}O$, $C_{10}H_{10}$, $C_{12}H_{16}$ and $C_{12}H_{20}O$ - all show a similar profile under illumination, they also show a different pattern in the dark. Specifically, a spike in emission was observed immediately after the lamps in the plant growth chamber were turned off. This phenomenon was observed for all light-to-dark transitions for these six BOCs, but was most prominent for $C_{10}H_{14}O$ (Fig. 4). An analogous event was previously observed for isoprene emitted from Sitka spruce, which was also accompanied by a spike in acetaldehyde (Hayward et al., 2004). Although acetaldehyde was not measured in this work, the lifetime of the spike in emissions is similar. Post-illumination acetaldehyde bursts have been observed for other tree species including oak and poplar (Li et al., 2010) and are proposed to originate from a biological process in which acetaldehyde is produced from an excess of pyruvate via a pyruvate overflow mechanism (Hayward et al., 2004).

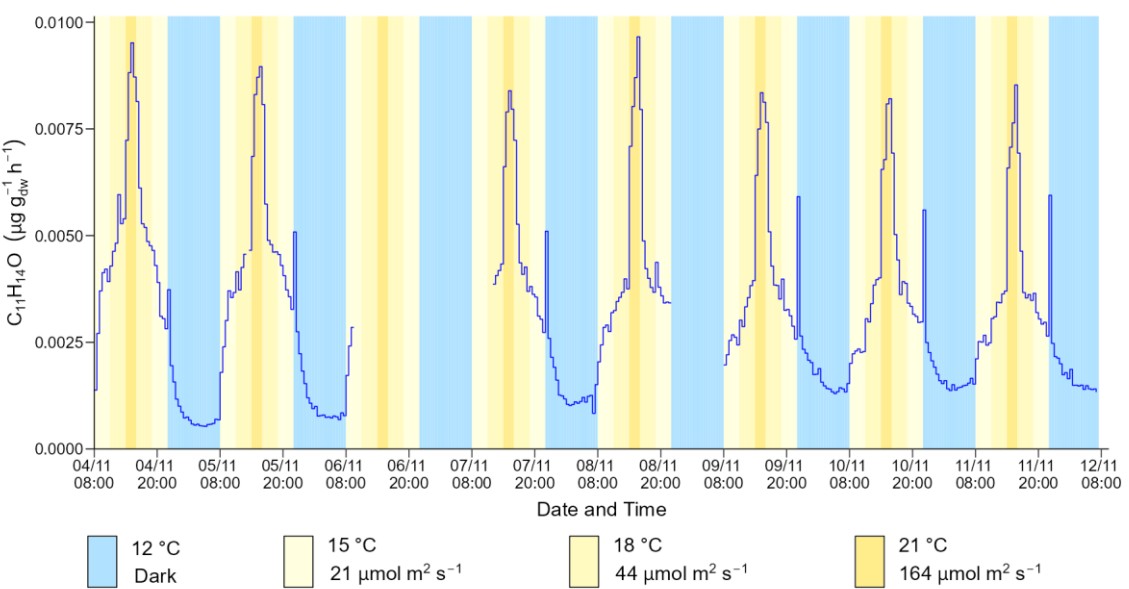

**Figure 4: Time series of the $C_{11}H_{14}O$ emission flux from Spruce 1 during the Daily Cycle.**

The BVOC emission patterns from Spruce 2 during the *Daily Cycle*, were not as well defined as for Spruce 1. As shown for the sum of monoterpenes in Fig. 5, the BVOC emission trend over a 24-hour period was similar to that from Spruce 1, with fluxes lowest in the dark, and peaking during the highest temperature and light conditions. For the intermediate conditions there was little difference observed in the emission fluxes. In addition, the fluxes did not always stabilise after a change in





conditions, often showing an increasing or decreasing trend. Due to the emissions not having a distinct pattern, the reproducibility across the week-long sampling period was quite poor.

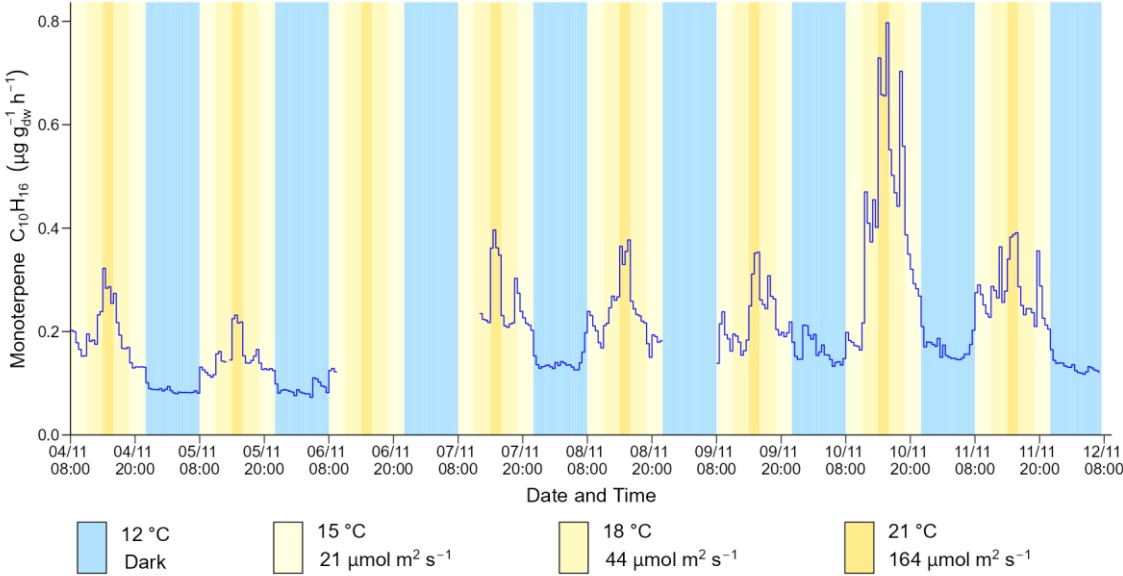

**Figure 5: Time series of the sum of monoterpenes ($C_{10}H_{16}$) emission flux from Spruce 2 during the Daily Cycle.**

The emission fluxes for the sum of monoterpenes and eight other BVOCs peaked strongly on 10/11/2021: the reasons for this are unknown. The conditions were the same as the other days of the *Daily cycle* and this unexpected rise in emissions was not observed for Spruce 1. For the majority of the BVOC emissions the magnitude of the flux increased as the *Daily Cycle* progressed and for a small number of BVOCs it decreased. A similar variation in emission fluxes was observed from stressed Norway spruce under controlled environmental conditions (Faiola et al., 2019) and none of the Spruce 1 BVOCs that exhibited post-illumination bursts were emitted from Spruce 2. These observations appear to provide further indication that Spruce 2 was stressed, and the normal biological activity of the tree was impaired.

### 3.3 BVOC emission pathways

### 3.3.1 Pooled emissions

For the *Temperature Cycle* the PPFD was held constant during the hours of illumination and the temperature was varied. The purpose of this cycle was to assess the contribution that temperature made to the emission flux, elucidating the importance of pooled BVOC emissions.





The average emission pattern of piperitone from Spruce 1 during the *Temperature Cycle* is shown in Fig. 6 (left), and is representative of almost all BVOC emissions from Spruce 1. Increasing the temperature from 12 °C to 15 °C in the dark caused a slight increase in the emission flux, indicating that the emissions were temperature dependent, and possibly originated from storage pools. In the dark, isoprene, $C_5H_8O_2$, $C_{10}H_{10}$, $C_{12}H_{16}$ and $C_{12}H_{21}O$ have very low emission fluxes and do not show any

variation with temperature indicating that they are unlikely to originate from storage pools. As with the *Daily Cycle* there was a spike in emission of these BVOCs following darkening and there was also a spike in the emission of $C_{11}H_{14}O$, although its emission profile had more similarities with that of piperitone.

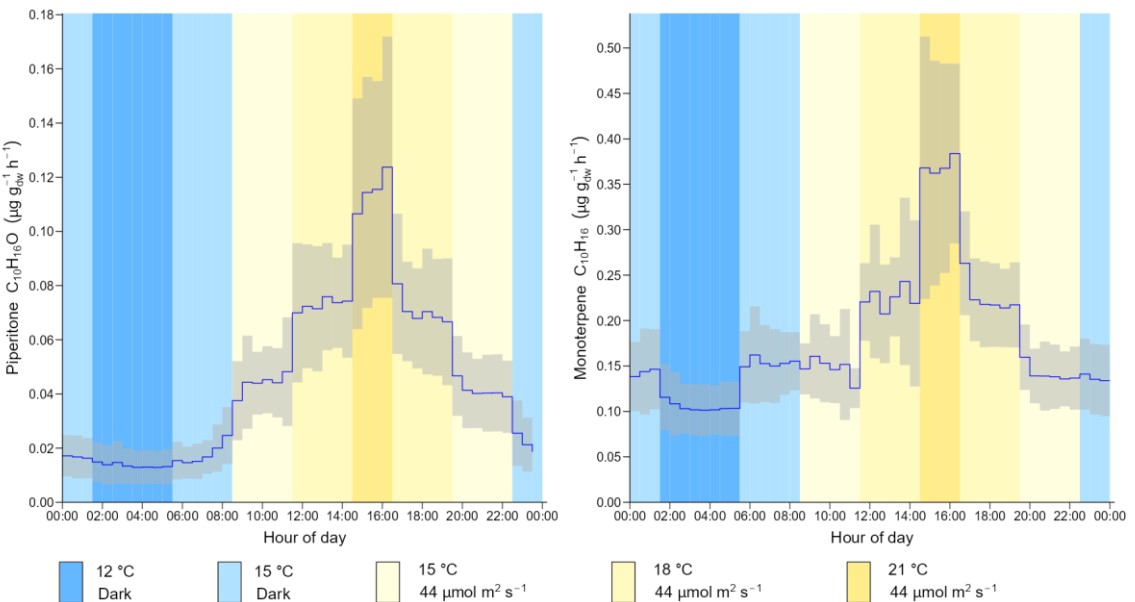

**Figure 6: Time series for the emission fluxes of piperitone from Spruce 1 (left) and the sum of monoterpenes Spruce 2 (right) during the Temperature Cycle. Blue line – measure data, grey shading – standard deviation.**

The time series for all Spruce 1 BVOCs show a significant increase in emission when the lamps were turned on, while the temperature was maintained at 15 °C, indicating light also plays a role in the emission of all BVOCs from Spruce 1. As the

temperature was increased during illumination the emission flux of all BVOCs increased, showing that the emissions were also temperature dependent. A temperature dependence has been observed for the emission of isoprene and monoterpenes from Sitka spruce previously (Street et al., 1996; Hayward et al., 2004). A correlation between oxygenated BVOCs and temperature has also been reported for sunflower (Schuh et al., 1997) and aspen (Ibrahim et al., 2010). For the majority of the BVOCs the change in emission flux with temperature was similar to that for piperitone, with the flux stabilising at each

temperature setting. Other BVOCs, such as isoprene, experienced a continual increase in emission flux upon illumination even though the temperature was maintained at 15 °C for three hours before and after illumination. This may be due to the emission of these BVOCs being closely linked to biosynthesis and the continual increase in emission is a result of the initiation of



biosynthesis until it reaches optimum efficiency. The emission of isoprene is assumed to occur immediately following biosynthesis (Kesselmeier and Staudt, 1999).


In contrast to Spruce 1, the BVOCs emitted from Spruce 2 had a much stronger response to changes in temperature, especially in the dark. Upon illumination there was a minimal change in the emission flux for all BVOCs. This is quite different to what was observed for Spruce 1, and indicates that the Spruce 2 BVOC emissions were more strongly influenced by temperature than those from Spruce 1. As observed for the *Daily Cycle* the data from Spruce 2 is quite scattered, and 20/11/2021 was the

only day with a clear BVOC emission profile. The BVOC emissions showed a pronounced response to changes in temperature and reached a stable emission flux immediately after each change. This indicates that the Spruce 2 BVOC emissions had a strong temperature dependence, and likely originated from storage pools (Kesselmeier and Staudt, 1999). The magnitude of the BVOC emission flux from Spruce 2 during the *Temperature Cycle* was lower than during the *Daily Cycle*. Again, this may have been a result of Spruce 2 suffering from stress which caused a reduction in the emission flux. It also indicates that the

BVOC emissions from Spruce 2 were largely due to emission from storage pools and emissions via the biosynthetic pathway were largely reduced due to stress.

### 3.3.2 Biosynthetic emissions

During the *Light Cycle* the temperature was maintained at 18 °C, and the PPFD was varied. The purpose of the *Light Cycle*
was to determine the impact of PPFD on the Sitka spruce emissions, and to evaluate the importance of the biosynthetic BVOC emission pathway.

Figure 7 shows the time series of piperitone emitted from Spruce 1 during the *Light Cycle*. All BVOC emission fluxes from Spruce 1 increased with PPFD during the *Light Cycle*. The variation in emission flux depended on the BVOC, with piperitone
showing a marked change in emission flux at each PPFD setting, while other emissions, such as $C_{10}H_{12}O_2$, only showed a significant increase in emission flux at 164 µmol m$^{-2}$ s$^{-1}$. The emission of isoprene from Sitka spruce has previously been shown to be PPFD dependent, with the emissions of monoterpenes from Sitka spruce reported to be PPFD independent (Evans et al., 1985; Hayward et al., 2004). This is in contrast to the results obtained here, where the sum of monoterpene emissions from Spruce 1 showed a clear response to PPFD changes. Monoterpene emissions are traditionally assumed to be temperature
dependent only (Tingey et al., 1980), although monoterpene emissions from Norway spruce (Van Meeningen et al., 2017) and oak (Kesselmeier et al., 1996) have been found to also have a light dependence. As with the previous cycles, a spike in emission following darkness was observed for isoprene, $C_5H_8O_2$, $C_{11}H_{14}O$, $C_{10}H_{10}$, $C_{12}H_{16}$ and $C_{12}H_{21}O$.



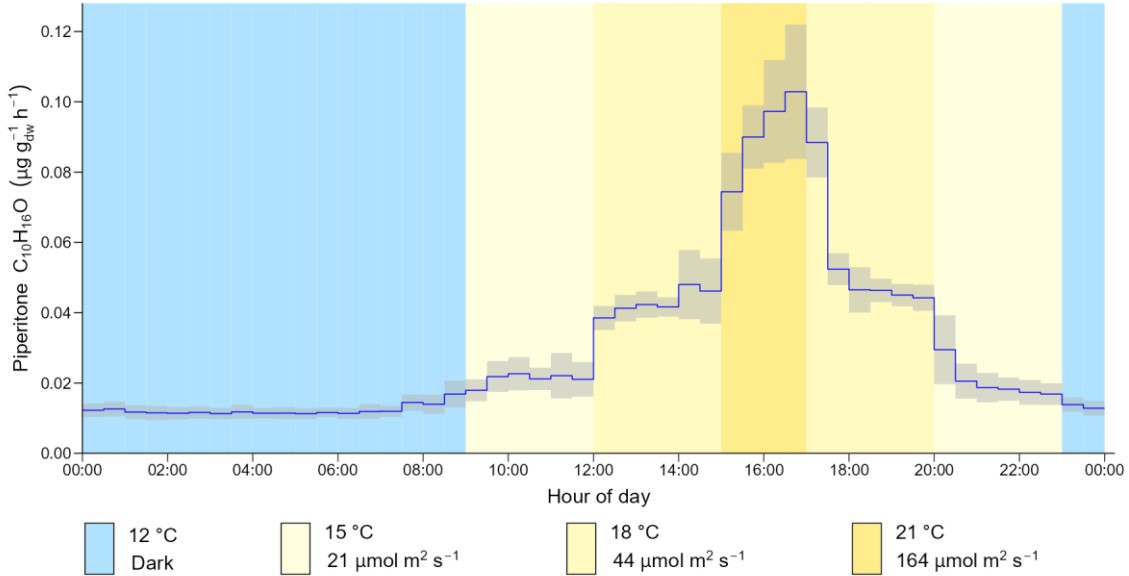

**Figure 7: Time series of piperitone flux from Spruce 1 during the Light Cycle. Blue line – measure data, grey shading – standard deviation.**

In contrast to Spruce 1, the emissions from Spruce 2 did not show a strong light dependence. Figure 8 shows the time series profile for the sum of monoterpenes during the *Light Cycle*, which is representative of most Spruce 2 emissions. There was no correlation between PPFD and the majority of the BVOCs emitted from Spruce 2, indicating that this tree was not sensitive to PPFD. The emission of monoterpenes[18], methanol and acetaldehyde[45] from Sitka spruce were previously found to be PPFD independent from 0 µmol m$^2$ s$^{-1}$ to 1000 µmol m$^2$ s$^{-1}$. Most BVOCs emitted by Spruce 2 did not follow any pattern, with the flux in the dark exceeding that under illumination on several occasions. It may be the case that the emission flux was quite low and near the limit of detection of the instrument or that the stress Spruce 2 had been suffering from was stronger during the *Light Cycle*, and obscured any potential emission patterns.



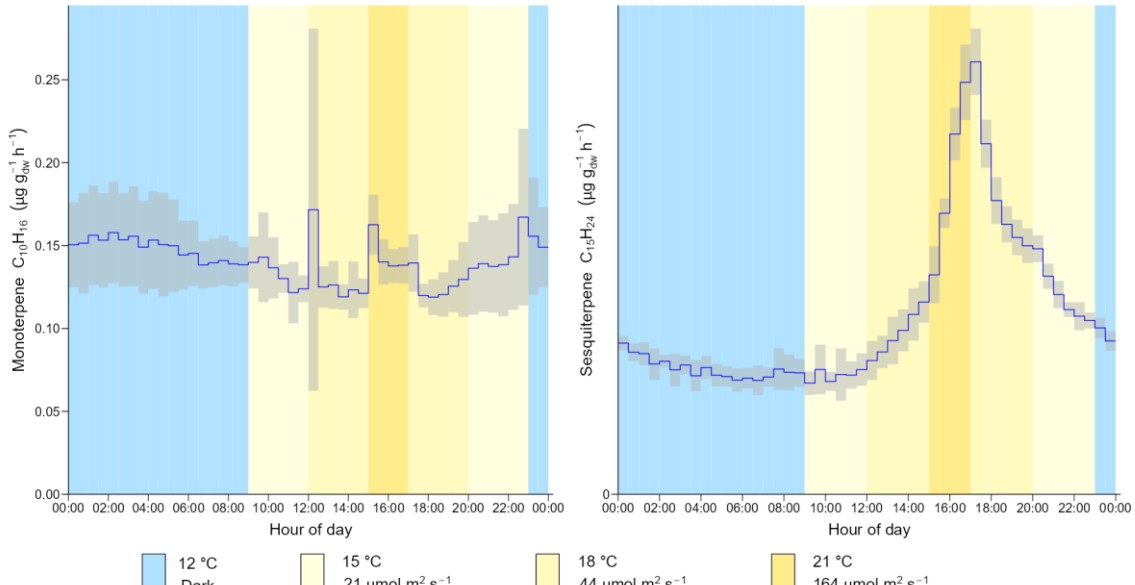

**Figure 8: Time series for the sum of monoterpene fluxes (left) and sum of sesquiterpene fluxes (right) from Spruce 2 during the Light Cycle. Blue line – measure data, grey shading – standard deviation.**


A light dependence was observed for a selected number of Spruce 2 emissions, including *E*-β-farnesene (Fig. 8) and camphor. *E*-β-farnesene showed the strongest PPFD dependence. This was unexpected as sesquiterpene emissions from Norway spruce were found to be PPFD independent (Bourtsoukidis et al., 2012). The change in *E*-β-farnesene flux was not very well correlated with the changes in PPFD, and did not stabilise at set values, but continually increased. There was also a time lag between the

changes in light intensity and emission flux, with the maximum flux often being recorded slightly after the light intensity had begun to decrease, and the increase occurring about an hour after illumination. The reason for the time delay is not clear and it was not observed for any other BVOCs. The other light dependent Spruce 2 emissions only showed a very small response to changes in PPFD, with most only responding at the highest PPFD of 164 µmol m$^{-2}$ s$^{-1}$.

**3.4 CO$_2$ fluxes**

**3.4.1 CO$_2$ flux dynamics**

The CO$_2$ flux was used to analyse the extent of photosynthesis and respiration over the course of the *Daily Cycle* (Fig. 9). For Spruce 1 the response of the CO$_2$ flux to changes in the PPFD was not as well defined as the BVOC emission flux. In general the CO$_2$ flux followed the expected trend and was highest in the dark, when photosynthesis and CO$_2$ uptake are not expected

to occur, and mitochondrial respiration takes place. A negative CO$_2$ flux was observed upon illumination, which became larger as the PPFD increased. This is an indication of photosynthetic activity and plant growth (Portillo-Estrada et al., 2020). For





Spruce 1 the flux in the dark varied between 0 nmol s$^{-1}$ g$^{-1}$ and 5 nmol s$^{-1}$ g$^{-1}$, similar to that previously observed for oak in the dark (Kesselmeier et al., 1996). At the highest PPFD (164 µmol m$^{-2}$ s$^{-1}$) and temperature (21 °C) the $CO_2$ flux was approximately 15 nmol s$^{-1}$ g$^{-1}$, which is comparable to the value of 20 nmol s$^{-1}$ g$^{-1}$ measured for Norway Spruce (Filella et al., 2007) at 23 °C and 200 µmol m$^{-2}$ s$^{-1}$. For Sitka spruce in the field (Brown et al., 1996) $CO_2$ fluxes were approximately 80 nmol s$^{-1}$ g$^{-1}$ at a minimum PPFD of 700 µmol m$^{-2}$ s$^{-1}$.

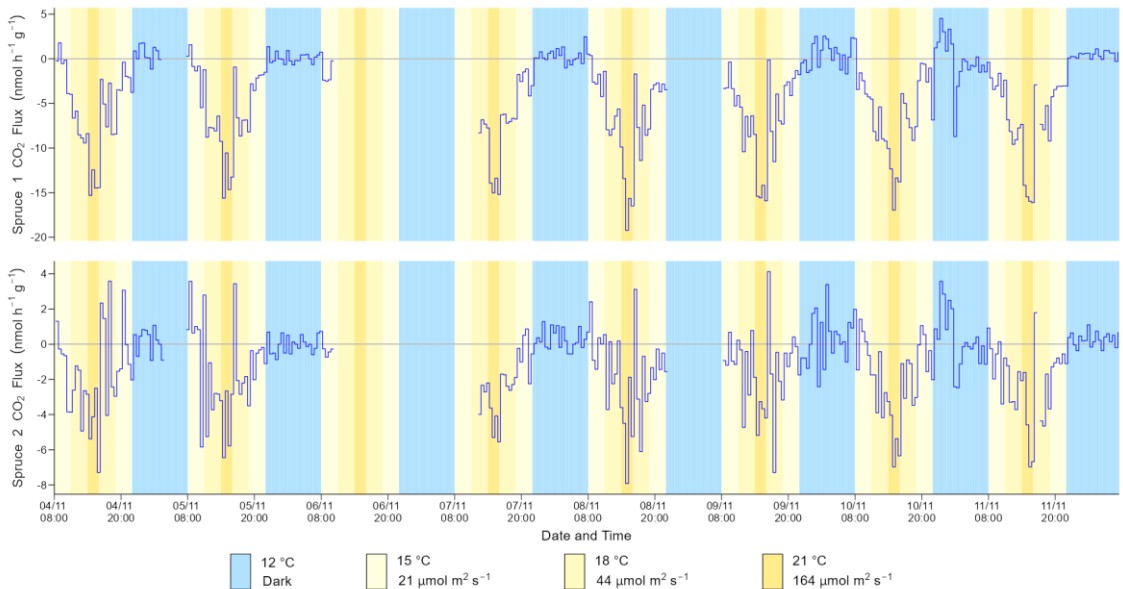

**Figure 9: Time series of $CO_2$ flux for Spruce 1 (top) and Spruce 2 (bottom) during the Daily Cycle.**

The $CO_2$ flux for Spruce 2 was quite scattered with spikes occurring regularly throughout the *Daily Cycle*. The trend was similar to that observed for Spruce 1, although the magnitude of the $CO_2$ flux was much lower, possibly because Spruce 2 was suffering from stress. However, while the photosynthetic $CO_2$ flux was reduced, there was no indication of chronic photoinhibition from the $F_v/F_m$ measurements (Table S2), which would have indicated stress-related damage to the photosynthetic apparatus. Instead, lower rates of photosynthetic $CO_2$ flux could have been caused by stomatal closure.

The $CO_2$ flux showed little dependence on temperature during the *Temperature Cycle* (Fig. S7). Upon illumination a significant decrease was observed, due to $CO_2$ uptake for photosynthesis. For Spruce 1, there was a slight decrease in the $CO_2$ flux with increasing temperature, suggesting photosynthetic activity was decreasing. An inverse relationship between photosynthesis and temperature has been observed for Sitka spruce previously (Neilson et al. 1972). This effect was not observed for Spruce 2.





The pattern of the $CO_2$ flux during the *Light Cycle* (Fig S8) was the same as that observed during the *Daily Cycle*, with a clear change in response to PPFD variations for both trees. The magnitude of the $CO_2$ flux during the *Light Cycle* was larger than

for the *Daily Cycle*. This is somewhat surprising as the PPFD was the same for both cycles, therefore the fluxes would be expected to be similar.

Across the three cycles, the Spruce 1 $CO_2$ flux was approximately five times higher than that for Spruce 2. The BVOC emission profiles from Spruce 2, indicated that the stress experienced by Spruce 2 was stronger during the *Light Cycle*, and was affecting

the physiology of the tree. On the other hand, the $CO_2$ uptake by Spruce 2, indicates that some biological processes were still functional. One possible explanation is that the physiological processes relating to biosynthetic BVOC emission are more sensitive to the stress experienced by Spruce 2 than photosynthetic processes.

### 3.4.2 Carbon balance

Net ecosystem carbon balance is an assessment of all inward and outward fluxes of carbon associated with an ecosystem (Bouvier-Brown et al., 2012). The Spruce 1 BVOC emission and $CO_2$ data for the *Daily Cycle* were used to calculate the carbon balance for Sitka spruce and to determine the contribution that BVOC emissions make to the overall carbon flux.

**Table 1: Contribution of $CO_2$ uptake, respiration and BVOC emissions to the carbon balance for Sitka spruce.**

| Carbon balance | Carbon flux | Respiration | BVOC emission |
|:---:|:---:|:---:|:---:|
| | mg C day$^{-1}$ g$^{-1}$ | % | % |
| Absolute uptake | - 4.07 | 6.8 | 0.2 |
| Respiration | 0.28 | - | - |
| BVOC emission | 0.01 | - | - |
| Net uptake | - 3.78 | 7.4 | 0.2 |


The absolute daily carbon uptake was calculated to be -4.07 mg C day$^{-1}$ g$^{-1}$, with an outward flux of $CO_2$ due to respiration of 0.28 mg C day$^{-1}$ g$^{-1}$, and a BVOC emission flux of 0.01 mg C day$^{-1}$ g$^{-1}$. This resulting in a net carbon uptake of -3.78 mg C day$^{-1}$ g$^{-1}$. The carbon loss due to BVOC emissions in terms of carbon units, was calculated to be approximately 0.2% of carbon taken in as $CO_2$. This is quite low and indicates that the contribution of BVOC emissions to the carbon balance for Sitka spruce

is relatively minor. For comparison, the emission of monoterpenes and isoprene were found to account for 0.93% of the net carbon flux from a Japanese larch plantation on an annual basis (Mochizuki et al., 2014). In another study the annual carbon loss from a Ponderosa pine forest due to BVOC emissions was calculated to be 0.6% on an absolute carbon scale, and 4% on a net carbon scale (Bouvier-Brown et al., 2012). BVOC emissions from a poplar plantation accounted for 0.63% of carbon





uptake, on a net carbon scale (Portillo-Estrada et al., 2018). The value calculated in this work is lower than that determined
from these field studies. However, the field studies were conducted over timescales of at least a year and over a range of
environmental conditions, and tree ages, both of which are expected to impact carbon uptake and BVOC emissions (Bouvier-
Brown et al., 2012; Tani and Mochizuk, 2021). The carbon balance for Sitka spruce determined here was calculated using
Spruce 1 *Daily Cycle* data and does not account for annual changes in environmental conditions or tree ages. Based on the
values determined from field measurements for other tree species, the exact contribution of BVOC emissions to the overall
carbon balance for Sitka spruce is expected to be greater under real-world conditions.

## 3.5 BVOC emission standardisation

BVOC emissions were standardised to a PPFD of 1000 µmol m$^{-2}$ s$^{-1}$, and a temperature of 30 °C, in line with recommendations
(Guenther et al., 1993). All BVOC emissions were subject to three standardisation procedures; pooled standardisation,
biosynthetic standardisation and combined standardisation. For each BVOC emitted from Spruce 1 and Spruce 2, the method
that best reproduced the emission profile for the *Daily Cycle* was selected and used to determine the standardised emission
flux (Fig. 9 and Table S7.1 and Table S7.2).

### 3.5.1 Standardisation procedures

**Pooled emission model**

Assuming the BVOCs were emitted from pooled sources, Eq. (6) was used to model the measurement data for the *Daily Cycle*
using $\beta = 0.09$ K$^{-1}$. However, the results did not match the measurement data from the *Daily Cycle*. To improve the model, a
unique $\beta$ value was calculated using Eq. (6) with the measurement data from the *Temperature Cycle* for each BVOC emitted
by each Stika spruce. These unique $\beta$ values were then used to calculate BVOC$_S$, the pool standardised emission flux for the
BVOC at 30 °C. The values for BVOC$_S$ and $\beta$ for each BVOC were used to model the data for the *Daily Cycle* for Spruce 1
and Spruce 2 (Fig. 10).





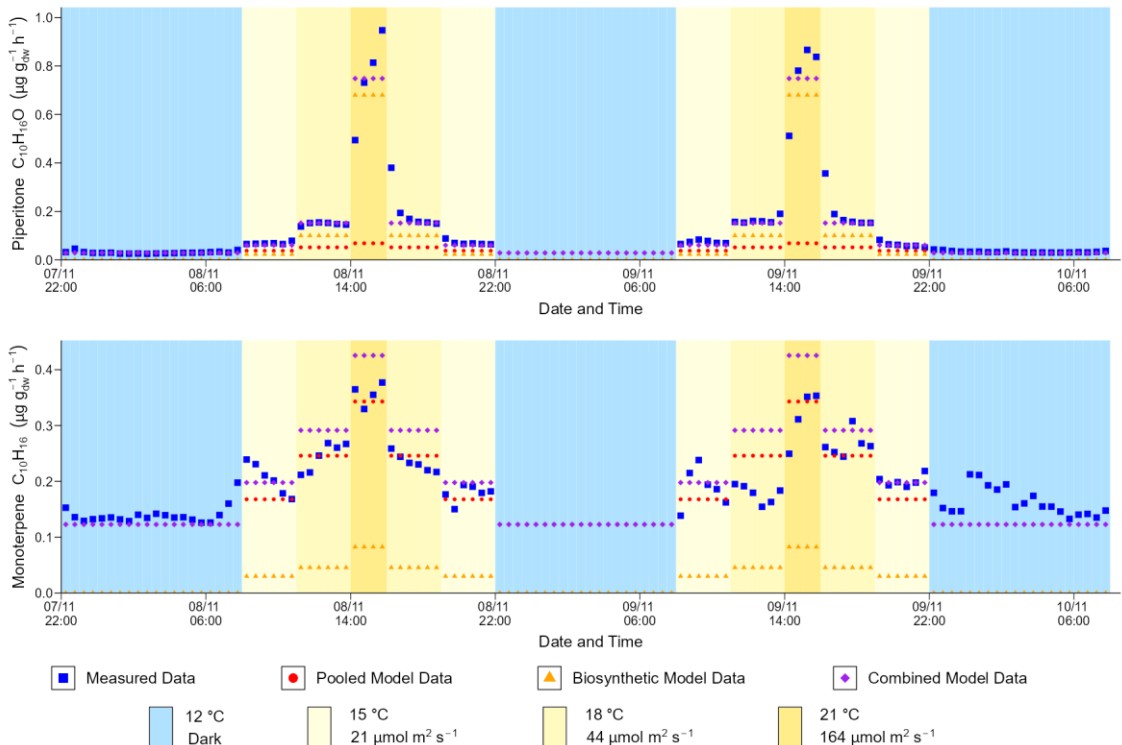

**Figure 10: Plot with measured and all modelled data for the emission of piperitone from Spruce 1 (top) and the sum of monoterpenes from Spruce 2 (bottom).**

For Spruce 1 the model was only capable of reproducing measurements recorded in the dark. Data acquired under illumination were significantly underestimated by the model for all BVOCs, particularly at the highest PPFD setting. This was unexpected, especially for monoterpenes which are assumed to be emitted solely from storage pools (Fuentes et al., 2000) and thus entirely temperature dependent with no PPFD dependence (Hayward et al., 2004). The emission of isoprene from Stika spruce has also been shown to be temperature and PPFD dependent (Evans et al., 1985), as it is emitted via a biosynthetic pathway, and therefore was not expected to be reproduced with the pooled emission model. The inability of the model to replicate the *Daily Cycle* measurements indicates that for all BVOCs emitted by Spruce 1 the biosynthetic pathways play a role in the emission process.

The emission profiles of six Spruce 2 BVOCs including, $C_6H_{10}O$ and $C_{20}H_{24}$, were completely described by the pooled emission model. For the remaining BVOCs the pooled model was unable to completely reproduce the measured data, indicating that biosynthetic pathways contributed to the emissions from Spruce 2 also. Among the sixteen BVOCs common to the emissions of Spruce 1 and Spruce 2, there was a large difference between the extent to which the pooled emission model could





reproduce the measured data. For example, 35% of the total emission flux of piperitone from Spruce 1 could be reproduced using the pooled emission model, while for Spruce 2 it accounted for 83% of the piperitone emission flux.

**Biosynthetic emission model**

Assuming the BVOCs originated from biosynthetic sources, Eq. (3) was used to model the measurement data for the *Daily*
*Cycle*. The temperature dependent term of the biosynthesis equation (Eq. (3)) was simplified with the removal of the denominator, as described previously by Schuh et al., (1997). It relies on $T_M$, the temperature of maximum enzyme activity, which was unknown for the Sitka spruce used in this study, but is expected to be close to 40 °C. Due to the large temperature difference anticipated between $T_M$ and the temperatures employed in this study, the denominator can be approximated to one. Unique coefficients were calculated for each BVOC from Spruce 1 and Spruce 2, and were used to model Spruce 1 and Spruce
2 *Daily Cycle* data (Fig. 10).

The biosynthetic emission model was much better than the pooled emission model at reproducing the Spruce 1 measurement data for the *Daily Cycle* and more accurately described the BVOC emissions under illumination. In total the biosynthetic model was able to completely reproduce the *Daily Cycle* emission profiles of 11 BVOCs emitted by Spruce 1, including isoprene.
This included five of the six BVOCs that underwent post-illumination spikes. For the remaining BVOCs emitted by Spruce 1 the biosynthetic model was not capable of completely describing the *Daily Cycle* emission profile, indicating that these BVOCs were also originating from storage pools.

The high correlation of the pooled emission model with Spruce 2 *Daily Cycle* data indicated that the contribution from the
biosynthetic emission pathway would be low, possibly due to inhibition of the biosynthetic pathways resulting from stress. This may explain why isoprene, a biosynthetic emission, was not detected in the emissions from Spruce 2, and six of the Spruce 2 BVOCs were completely replicated by the pooled emission model. The sum of monoterpene signal did show a correlation with the biosynthetic emission model, although it could not fully explain the emission flux. Previous studies involving Sitka spruce have not reported a biosynthetic factor in monoterpene emissions (Hayward et al., 2004), but it has been observed for
beech (Schuh et al., 1997) and holm oak (Lang-Yona et al., 2010) previously.

**Combined emission model**

The emission fluxes of eleven BVOCs from Spruce 1 and six BVOCs from Spruce 2 were completely reproduced using the biosynthetic emission model and pooled emission model, respectively. All other BVOCs displayed contributions from both
emission pathways. A combined biosynthesis and pooled emission standardisation equation developed by Schuh et al. (1997) (Eq. 7) was used to standardise and model the emission fluxes.



In total, the emission fluxes for 23 BVOCs emitted from Spruce 1 were reproduced by the combined emission model. The contribution each pathway made to the total emission was BVOC dependent, although for all BVOCs the contribution from the biosynthetic pathway was larger. The combined emission model was able to accurately describe the remaining 26 BVOC emissions from Spruce 2, including the sum of monoterpenes. This is the first time that the combined emission pathway has been used to standardise the emissions of Sitka spruce as previous studies used only the pooled emission model for monoterpenes (Hayward et al., 2004), or biosynthetic model for isoprene (Street et al., 1996). The combined emission pathway has been used previously to describe the emissions from sunflower and beech (Schuh et al., 1997) as well as Norway spruce (Laurila et al., 1999).

The combined emission equation was used to standardise 13 of the 16 emissions common to both Spruce trees. The remaining three BVOCs: $C_9H_8O$, $C_{10}H_{14}O_2$ and $C_{20}H_{24}$, originated from different pathways in each tree. The emission of $C_9H_8O$, identified as *E*-cinnamaldehyde, from Spruce 1 originated from a biosynthetic pathway, while from Spruce 2 it appeared as a pooled emission. For $C_{10}H_{14}O_2$ and $C_{20}H_{24}$ the emission from Spruce 1 was via both pathways, while for Spruce 2 the emission was via the pooled pathway only. This is consistent with other observations which suggest that both pathways are open for a healthy tree (Spruce 1), but when a tree suffers stress (Spruce 2), the biosynthetic emission pathway is closed and the only available emission pathway is diffusion from storage pools.

### 3.5.2 Standardisation application

The modelling approach which best replicated the *Daily Cycle* BVOC emission profile for each Sitka spruce tree was used to provide the standardised emission flux at a PPFD of 1000 µmol m$^{-2}$ s$^{-1}$ and temperature of 30 °C. The standardised emission fluxes for Spruce 1 are presented in Table S6 and for Spruce 2 in Table S7.

Piperitone was the dominant BVOC emitted by Spruce 1, with a standardised emission flux of 17.29 µg g$_{dw}^{-1}$ h$^{-1}$. Although piperitone has been detected in the emissions of Sitka spruce previously, an emission flux was not provided (Tani et al., 2003). The standardised emission flux of isoprene from Spruce 1 was 6.3 µg g$_{dw}^{-1}$ h$^{-1}$. These values are within the range of previously reported values determined under a variety of environmental conditions, Table 2.





**Table 2: Reported emission fluxes for isoprene and monoterpenes from Sitka spruce.**

| Isoprene | Monoterpene | Temperature | PPFD | Reference |
|---|---|---|---|---|
| $\mu g\ g_{dw}^{-1}\ h^{-1}$ | $\mu g\ g_{dw}^{-1}\ h^{-1}$ | °C | $\mu mol\ m^{-2}\ s^{-1}$ | |
| 3.06 | 0.95 | 28 | 1000 | Evans et al., (1982) |
| 5.21 | 2.30 | 38 | 1000 | Evans et al., (1985) |
| 0.05 | 1.50 | 16 | 360 | Street et al., (1993b) |
| 1 - 5 | 0.001 – 0.005 | 21 - 36 | not reported | Street et al., (1996) |
| 14 – 17 | - | 28 | 230 | Hayward et al., (2002) |
| 13.4 | 2.97 | 30 | 1000 | *Hayward et al., (2004) |
| 6.3 | 0.93 | 30 | 1000 | *This work |

*Emission fluxes standardised, otherwise reported at measured conditions.*

The standardised emission fluxes of most BVOCs emitted by Spruce 2 were lower than those for Spruce 1. As mentioned
above, this is likely due to Spruce 2 suffering from some unknown stress which lowered the effectiveness of the biosynthetic
emission pathway. The standardised total monoterpene emission flux from Spruce 2 was 1.2 $\mu g\ g_{dw}^{-1}\ h^{-1}$, higher than the flux
determined for Spruce 1. In this case, a direct comparison with previous studies on stressed Sitka spruce is not possible,
although it is worth noting that increased monoterpene emissions have been observed from blue spruce (Faiola et al., 2015)
and Norway spruce (Bourtsoukidis et al., 2012) under stressed conditions.

The emission flux data obtained for piperitone, isoprene and the sum of monoterpenes from Spruce 1, have been scaled to
provide an estimate of the annual emission fluxes for these BVOCs from all Sitka spruce in Ireland. The annual emission
fluxes for the three main BVOCs emitted from Spruce 1, piperitone, isoprene and the sum of monoterpenes, were calculated
using Eq. (8):

$$BVOC_{flux} = E_{BVOC} \times LMA \times Area, \tag{8}$$

where *BVOC*$_{flux}$ is the annual emission of a particular BVOC from all Sitka spruce in Ireland, in tonne year$^{-1}$; *E*$_{BVOC}$ is the
annual emission flux of a particular BVOC per dried mass of Sitka spruce, in tonne year$^{-1}$ g$^{-1}$; *LMA* is foliar mass per ground
area of Sitka spruce, in g m$^{-2}$; and *Area* is the total land area of Sitka spruce forests in Ireland, in m$^2$. *E*$_{BVOC}$ was calculated for
each BVOC using the standardised emission flux equations combined with global solar radiation and temperature data acquired
from Met Éireann (Kelly., 2023; Eireann). An estimate for LMA was derived from assessments of Sitka spruce forests in
Ireland (Tobin et al., 2006) and the UK (Barton and Jarvis, 1999; Meir et al., 2002; Norman and Jarvis, 1974; Dengel et al.,
2015). The calculated annual BVOC emission fluxes for piperitone, isoprene and monoterpenes are given in Table 3.





**Table 3: Table of calculated annual emission fluxes for piperitone, isoprene and monoterpenes from Sitka spruce in Ireland.**

| BVOC | Annual emission flux | | |
|---|---|---|---|
| | tonne year$^{-1}$ | g C ha$^{-1}$ year$^{-1}$ | g C m$^{-2}$ year$^{-1}$ |
| isoprene | 13,000 | 34,000 | 3.3 |
| piperitone | 8,200 | 19,000 | 1.9 |
| monoterpenes | 1,600 | 4,200 | 0.4 |
| total | 22,800 | 57,200 | 5.6 |

The total annual emission flux for the three main BVOCs emitted from all Sitka spruce in Ireland was calculated to be 22,800

tonne year$^{-1}$. This figure is proposed as a lower estimate of the true emission flux for several reasons. Firstly, due to limited data availability, the calculation does not account for increases in foliage (affecting $E_{BVOC}$ and $LMA$) as the trees grow (Bouvier-Brown et al., 2012). Secondly, the calculation only accounts for the three main foliar BVOC emissions and does not include emissions that may originate from the bark of the tree. As previously discussed, visible resin deposits were observed on the trees in this study and potentially contributed to the overall emission flux. Finally, the calculation does not take all BVOCs

emitted from Sitka spruce into consideration.

The total non-methane VOC emissions produced by anthropogenic sources in Ireland for 2020 was 112,600 tonne year$^{-1}$, as reported by the Environmental Protection Agency (P. Duffy, 2022). This figure is based on bottom up emissions inventory calculations for air pollutants resulting from human activity, and does not account for BVOC emissions (Agency, 2019). This

implies that in Ireland the total emission of BVOCs from Sitka spruce is equivalent to 20% of the total anthropogenic VOC flux, and shows that biogenic sources account for a significant proportion of ambient VOCs released into the atmosphere.

An annual BVOC emission flux has not been previously calculated for a Sitka spruce forest, although the value calculated in this work falls within the range of previously reported values for other plantation tree species. An annual BVOC emission flux

of 19,200 g C ha$^{-1}$ year$^{-1}$ was reported for a poplar plantation in Belgium (Portillo-Estrada et al., 2018), three times lower than the 56,820 g C ha$^{-1}$ year$^{-1}$ calculated for Sitka spruce in this work. The emissions from the poplar plantation were dominated by isoprene, methanol and acetic acid, all of which are low in carbon number, while in this work piperitone and monoterpenes, both $C_{10}$ BVOCs, made substantial contributions to the emission flux. BVOCs with a higher carbon number will contribute more if the emission is reported in terms of carbon mass, and this may partially explain why the emission flux is higher for

Sitka spruce. An annual BVOC emission flux of 9.4 g C m$^{-2}$ year$^{-1}$ was calculated for a Ponderosa pine forest in California





(Bouvier-Brown et al., 2012). This is higher than the 5.7 g C m⁻² year⁻¹ calculated in this work for Sitka spruce. Each tree and plant species emits a unique BVOC profile, which will lead to differing values for annual emission fluxes.

The annual emission fluxes of piperitone, isoprene and monoterpenes from Stika spruce in Ireland were calculated to be 8,240 tonne year⁻¹, 13,115 tonne year⁻¹, and 1620 tonne year⁻¹, respectively. The annual emission of isoprene is greater than the annual emission for piperitone, despite piperitone being identified as the dominant emission in this work, and having a standardised emission flux almost three times that of isoprene. This is due to the relationships between temperature, PPFD and BVOC emission, which are different for piperitone and isoprene (Fig. 11).

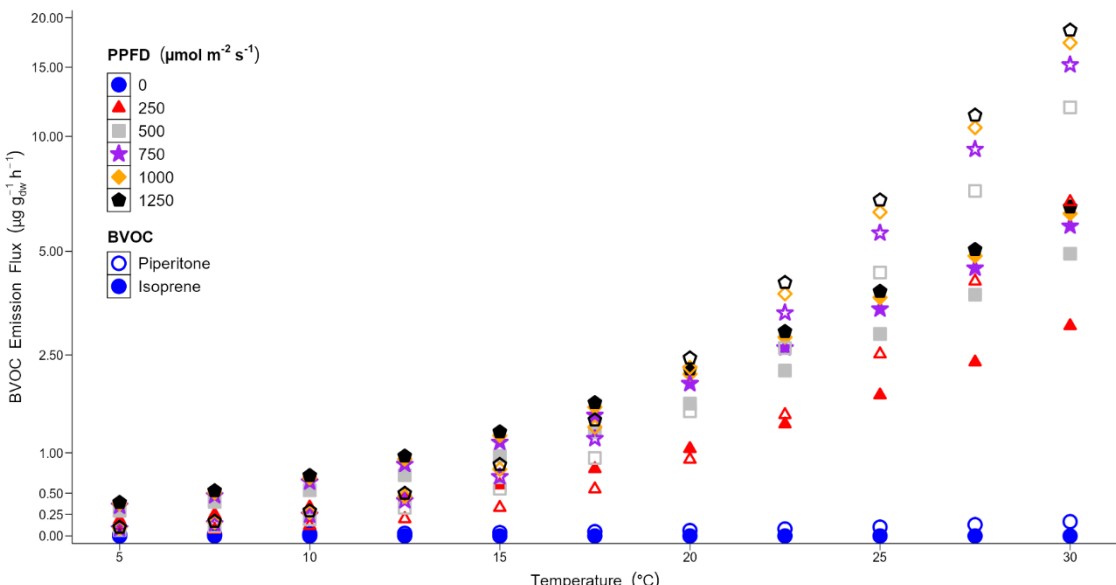

**Figure 11: Relationships between piperitone and isoprene emission fluxes with temperature at different PPFDs. Colour and shape indicate different PPFD, empty symbols – piperitone, filled symbols – isoprene.**

BVOC emission fluxes have an exponential relationship with temperature. Above a certain temperature, there is an inflection point where the emission flux increases significantly with increasing temperature. As shown in Fig. 11 piperitone (open symbols) has a strong exponential relationship with temperature, with an inflection point around 15 °C. Below this temperature, the emission is quite low and there is little change in piperitone emission with temperature. Isoprene (closed symbols) has a much weaker relationship with temperature than piperitone, with an inflection point at around 22.5 °C, although it is not as clearly defined as it is for piperitone.

The emission fluxes for both BVOCs have a logarithmic relationship with PPFD. At a given temperature, the increase in emission flux decreases with fixed successive increases in PPFD, until a saturation point is reached. Below 750 μmol m² s⁻¹





the piperitone emission flux is strongly PPFD dependent. This relationship weakens as PPFD increases and the emission flux approaches the saturation point. The isoprene emission flux has a stronger relationship at lower PPFD than piperitone. This is seen in Fig. 11 through the sharp increase in emission flux between 0 and 250 µmol m$^2$ s$^{-1}$. The saturation point for isoprene is lower than for piperitone. The change in isoprene emission flux above 500 µmol m$^2$ s$^{-1}$ is significantly less than at lower PPFD values.

The exponential temperature and logarithmic PPFD relationships operate simultaneously. For both piperitone and isoprene, at lower temperatures the influence of PPFD is stronger than that of temperature. Above the respective inflection points the temperature factor plays a greater role in the emission flux. At lower temperatures the emission flux of piperitone is less than that for isoprene. This is because of the strong exponential relationship between the piperitone emission flux and temperature. Isoprene has a more muted relationship with temperature and the emission flux is not as sensitive to lower temperatures. In the lower temperature regime the emission of isoprene is higher than piperitone, at all PPFD levels.

Typical temperatures for Ireland range between 6 °C and 17 °C. Under these conditions the emission of isoprene is higher than piperitone. The temperature for standardising BVOC emissions is 30 °C. At this temperature the emission of piperitone is higher than isoprene. The conditions employed during the measurement cycles in this work fall between these two extremes, and favour the emission of piperitone. Analysis of the temperature and PPFD dependence of the emission of both BVOCs explains why the annual emission flux of isoprene is greater than that for piperitone. The predicted increase in surface temperatures due to climate change (IPCC, 2023), would be sufficient to alter the piperitone–isoprene emission ratio from Sitka spruce in favour of piperitone. Therefore, under current future climate warming projections, piperitone would be the dominant BVOC emitted by Sitka spruce.

**4 Conclusions**

The emissions from Sitka spruce were found to contain 74 different BVOCs, 52 of which were oxygenated. For healthy Sitka spruce, piperitone was the dominant emission, with a standardised emission flux of 17.29 µg g$_{dw}^{-1}$ h$^{-1}$. The second highest emission was isoprene, with a standardised emission flux of 6.3 µg g$_{dw}^{-1}$ h$^{-1}$. Five monoterpenes were identified; myrcene, β-phellandrene, δ-limonene, α-pinene, and camphene. The standardised emission flux for the sum of all monoterpenes was 0.93 µg g$_{dw}^{-1}$ h$^{-1}$. Six of the BVOC emissions, including isoprene, experienced a post-illumination burst. BVOC emissions accounted for approximately 0.2 % of carbon uptake.

One Sitka spruce tree was suffering from an unknown stress, which was reflected in the BVOC and CO$_2$ measurements. The BVOC fluxes differed from those determined for the healthy Sitka spruce, and were dominated by monoterpenes, with a



different composition ratio and a standardised emission flux of 1.3 μg $g_{dw}^{-1}$ $h^{-1}$. The emissions also contained a sesquiterpene and six $C_6$ oxygenated BVOCs.

The dependence of the BVOC fluxes on temperature and PPFD was investigated using three different measurement cycles. The emissions from the healthy spruce were found to respond to changes in temperature and PPFD and were found to originate from pooled and biosynthetic emissions pathways. In contrast, the emissions from the stressed spruce were found have a strong correlation with temperature and minimal PPFD dependence indicating the BVOCs were primarily emitted from pooled sources.

The BVOC emissions were standardised to a temperature of 30 °C and PPFD of 1000 μmol $m^{-2}$ $s^{-1}$. The majority of the BVOC emission fluxes were standardised using a combined biosynthetic and pooled emission model, indicating that the emissions from Sitka spruce originate from both pooled and biosynthetic emission sources simultaneously.

The standardised emission fluxes for piperitone, isoprene and the sum of monoterpenes from the healthy Sitka spruce were used to extrapolate annual fluxes for all Sitka spruce plantations in Ireland. The annual emission flux of piperitone was found to be 8,240 tonne $year^{-1}$, for isoprene 13,114 tonne $year^{-1}$, and for the sum of monoterpenes 1,618 tonne $year^{-1}$. The annual emission flux for isoprene was higher than that for piperitone, and is related to their temperature and PPFD dependences. Under the conditions of the Irish climate the emission flux of piperitone is lower than isoprene. This highlights the importance of assessing BVOC emissions at the conditions relevant to the local climate.

**Code and data availability**

Code and data are available at: 10.5281/zenodo.10514476

**Supplement link:**

Supplement is available at:



**Author contribution**

HF, JK, JW and AW designed the experiments, which were carried out by HF. IS, DM and KK performed TD-GC/MS analysis of samples and HF did all other analysis in consultation with JK, AW and JW. HF prepared the manuscript with contributions from all authors.

**Competing interests**

The authors declare that they have no conflict of interest.

**Acknowledgements**

This work was funded by the Irish Environmental Protection Agency through an Irish Research Council, Government of Ireland Postgraduate Scholarship (GOIPG/2019/1189) and by the European Union's Horizon 2020 research and innovation
programme (EUROCHAMP-2020, grant no. 73097).

The authors would like to thank Dr Dean Venables from University College Cork for laboratory access.

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
