# Peer review of "Identification of volatile organic compounds emitted by Sitka spruce and determination of their emission pathways and fluxes"

_EGUsphere, 2024_

## Author Comment (AC1)

**Reply to Referees' Comments**

*The authors would like to thank the referee for the detailed review of the manuscript and the helpful comments that will help us to provide an improved version of the manuscript. We address all the comments below and we will further integrate the corresponding changes in a revised article.*

**Reviewer 1**

**General comments:**

The authors investigated the BVOC emissions from Sitka spruce under laboratory conditions. Sitka spruce is an important tree species and is commonly used as part of afforestation programs. Using a combination of TOF-CIMS and TD-GC-MS techniques, the authors were able to detect many compounds that have not previously been reported in the emissions from Sitka spruce. The authors also investigated the temperature/PPFD response and emission pathways of the different BVOCs to determine if they originated from storage pools, biosynthesis, or both. The authors then extrapolated their results to estimate the annual BVOC emission fluxes for all Sitka spruce plantations in Ireland.

The main weakness of this study is the low number of plant replicates used. Only 3 seedlings were sampled in this study, and one of them (Spruce 3) was unhealthy and its emissions are not reported. Furthermore, there is a lot of variability in the BVOC emission profiles, temperature/PPFD response, and $CO_2$ fluxes of the two remaining seedlings (Spruce 1 and 2). For example, isoprene was the second highest emission from Spruce 1, but was only emitted at trace levels from Spruce 2. The authors also note that Spruce 2 might have experienced mild stress, which may be why its emission behaviour was different from Spruce 1. It is hard to obtain meaningful/convincing statistics and to draw generalized conclusions about the emission behaviour of Sitka spruce from these results. I would strongly recommend that the authors conduct measurements on additional Sitka spruce seedlings. There should be at least 3 healthy plants to obtain meaningful statistics and to account for intraspecific variability. There is only one healthy plant (Spruce 1) and one mildly-stressed plant (Spruce 2) in the current study. And if the authors wish to compare the emission behaviour of healthy vs. stressed Sitka spruces, then ideally, they should include 3 stressed plants in addition to the 3 healthy plants.

*Author reply (AR): The authors understand the referee's point of view because this study initially intended to replicate all measurements on three healthy plants. Although the aim of the study was not fully realised, we do believe that the data included in the present paper are robust and provide valuable new information for the scientific community that has not been published elsewhere. In particular, the authors would like to highlight the following aspects of this work:*

- *The novel experimental set up combining complementary online ToF-CIMS and offline TD-GC-MS for non-target analysis of the emissions.*
- *Measurements performed for seven days on diurnal cycles for the three modalities (temperature, light, and daily cycles) to make the results robust.*
- *Identification of 74 BVOCs in the emissions, while less than 10 were reported in previous studies.*
- *Piperitone identified as the main emitted BVOC while it was only reported as a minor emission previously*
- *First study to obtain a comprehensive understanding of the emissions pathways (pooled and de novo) for Sitka spruce*
- *New evidence to show that the BVOC emissions are best modelled using a combination of pooled and de novo pathways. This combined model is currently not included in large scale models.*

*Considering this study represents a very large amount of experimental work and data analysis, it is impossible to conduct additional measurements on other spruce plants within a short timeframe. In addition, it would not be possible to find spruce saplings with the same genetic origin as the ones used in the present study, which would introduce a bias. However, we accept that using emissions from one healthy tree to provide estimates of emissions for scaling up purposes and carbon balance is quite uncertain. As a result, we will significantly reduce this part of the article in a revised manuscript.*

**Line-by-line comments:**

L79: Since these are not mature trees, I would suggest replacing the word "tree" with "seedling". Please also provide the height of the 3 seedlings used in this study.

*AR: These are young trees with a height of 60 - 75 cm. We will replace the word "tree" by "sapling", which we believe to be more appropriate than "seedling".*

L89: Please provide the manufacturer and model of the lamps, if available.

*AR: The lamps were Panasonic FL40SSW/37-PRF3 Fluorescent Lamps (added to the manuscript).*

L90: PPFD is usually expressed in units of $\mu mol\ m^{-2}\ s^{-1}$. Please correct this.

*AR: This will be corrected in the revised manuscript.*

Figure 1: Recommend replacing "Ball Meter" with "Ball Flowmeter" to make its purpose clearer to the reader.

*AR: This will be corrected in the revised manuscript.*

L99: Why is there such a large difference in the flow rates into each enclosure? Large differences in flow rates can result in different humidity levels in each enclosure, which in turn may affect the BVOC emission rates. For future experiments, it would be advisable to try to equalize the flow rates into each enclosure using mass flow controllers (or more cost-effective needle valves).

*AR: Thank you for the comment. Firstly, the original manuscript contained an error – the flow did not vary between 1.5 and 7 L min$^{-1}$, but between 4.5 and 7 L min$^{-1}$. This will be corrected in the revised manuscript. Secondly, we agree that it is better to ensure the same flow when enclosures are identical, but in our case, the volume of the enclosures surrounding the saplings was slightly different. Consequently, we adjusted the flow through the enclosures to compensate for the volumetric variations and to keep the residence time as similar as possible between different enclosures.*

L110: The relative humidity inside the miniature enclosure may not be the same as the RH in the other enclosures, as the RH would be dependent on the amount of foliage in each enclosure, the flow rate into the enclosure, and the transpiration rate of the enclosed plant.

*AR: True, the relative humidity was probably a little bit higher in tree enclosures, but it is not expected to affect BVOC emissions, as to our no knowledge there is no study showing impact of air humidity on BVOC emissions. In addition we did not observe and condensation on the enclosure walls, which would favour wall deposition, especially for oxygenated BVOCs*

Section 2.2: Please state the detection limit and accuracy (measurement uncertainty) of the TOF-CIMS and TD-GC-MS.

*AR: Regarding TD-GC-MS, standards were used for qualitative identification purposes only. For ToF-CIMS measurements, the detection limit and accuracy is different for each VOC, but based on the background and calibration tests, the detection limit is around 10s of ppt for monoterpenes, with a measurement uncertainty typically at 30%, in agreement with a previous study using benzene ToF-CIMS (Lavi et al., 2018). To identify an emitted VOC, we used a statistical test (Welch's t-test) to select compounds with ion signals that are 3 times higher than the empty enclosure, meaning that we did not focus on ions that were only present in very small amounts. Taking this into consideration, only compounds with concentrations higher than ca. 0.1 ppb in the enclosures were analysed.*

L155: Just to confirm, are you sure that the heated transfer line was connected to a split/splitless injector, and not directly to the GC column?

*AR: The transfer line was directly connected to the GC column. This will be amended in the revised manuscript.*

L159: The phrase "fire purge" is a bit misleading. I think you mean to say "pre-trap fire purge", or "pre-desorption purge".

*AR: Thanks. The correct terminology is indeed pre-trap fire purge. This will be corrected in the revised manuscript.*

L162: You listed the dimensions of the GC column as 60 m × 0.3 mm × 1.8 μm. 0.3 mm is not a standard column ID. Please specify whether the column inner diameter (ID) was 0.25 mm or 0.32 mm.

*AR: The column inner diameter was 0.32 mm. This will be added in the revised manuscript.*

L163: For clarity, please specify if the pressure was 23 psig or 23 psia.

*AR: The pressure was 23 psig. This will be added in the revised manuscript.*

L209-210: These PPFD levels seem quite low. For comparison, can you provide the typical PPFD levels expected in Irish summer conditions?

*AR: It is true that these PPFD levels were rather low, but they are representative of the summer daily average PPFD in Ireland, as indicated by Met Éireann, the national meteorological organisation in Ireland. However, peak PPFD levels can sometimes rise to 1000 μmol m$^{-2}$ s$^{-1}$, see for example figure 1 in Byrne et al. (2005).*

L229: How were the Tenax tubes sealed? Did you use Swagelok brass caps?

*AR: The Tenax tubes were sealed using Swagelock brass caps fitted with Teflon ferules, which were first sealed by hand and then tightened further using the tool supplied by the manufacturer. This clarification will be added to the revised manuscript.*

L232: Is it possible that the plant biomass might have changed during the BVOC measurement period?

*AR: Yes, the plants grew slightly during the measurement period, but we believe that the biomass did not increase significantly (compared to total biomass enclosed). Any changes would be hard to quantify, as dry biomass weighing is a destructive method.*

L243: The "Flow" term is missing from Eq. 1. Please correct this.

*AR: This will be corrected in the revised manuscript.*

L278: Some monoterpenes originate from de novo biosynthesis and show both temperature- and light-dependency. For example, see study by Ghirardo et al. (2010) on Norway spruce and other species:

https://doi.org/10.1111/j.1365-3040.2009.02104.x

*AR: Thank you for this comment, we will add this detail in the revised paper with the reference* Ghirardo et al. (2010)

Section 3.1: Can you include a figure that shows the relative contributions of individual compounds to the total BVOC emissions from Spruce 1 and Spruce 2? For example, you mention that myrcene was the dominant monoterpene emitted from both spruces, followed by β-phellandrene, but this is not illustrated in any of your figures/tables.

*AR: Thank you for this comment, it is true that this information was not included in a plot in the original manuscript, even if this can be inferred from the table of standardisation emission fluxes in the SI (BVOCs are listed in order of decreasing standardised BVOC emission flux). We will insert a figure in the Supplementary Information to support our statement, in which boxplots representing median emissions and quantiles (5, 25 75 and 95) of each individual VOC are represented (see figure below). Identification of the main isomers (for example, myrcene as the dominant monoterpene) was inferred from chromatogram peak areas in the TD-GC-MS data. This information is available in the dataset associated with the paper.*

[Figure]

*Box plot for Spruce 1 (top) and Spruce 2 (bottom) BVOC emissions (in µg $g_{dw}^{-1}$ $h^{-1}$). The bar represents the median, the bottom and top limits of the box are the 25th and 75th quantiles, and the end of the bottom and top bars represent the 5th and 95th quantiles.*

L392: Can you specify which BVOC species showed a pronounced increase in emission upon illumination, and which had a more muted response?

*AR: Most of the BVOC emissions showed a pronounced increase upon illumination. The ones which had a more muted response are: $C_5H_{10}O_6$, $C_9H_8O$, $C_{12}H_{20}O$, $C_{10}H_{14}O$, $C_7H_{12}N_2$, $C_6H_{14}O_3$, $C_{12}H_{14}$, $C_{10}H_{10}$, $C_{20}H_{24}O_2$, $C_{10}H_{20}O_2$, $C_3H_8N_2O$, $C_{10}H_{18}O_4$, $C_7H_4O_4$, $C_{18}H_{34}O$. We can add this specific information to the manuscript.*

L402: Is it $C_{10}H_{14}O$ or $C_{11}H_{14}O$?

*AR: Thanks. It is $C_{11}H_{14}O$ and will be corrected in the revised manuscript.*

L404: What is the average lifetime of the post-illumination bursts observed in your study?

*AR: In this work, we used a multivalve to switch between each sapling every 7.5 minutes, which did not allow for identifying the exact timing of the spike in emissions. As a consequence, the post-illumination burst was only observed during a single measurement, indicating that it was a short event (less than 30 minutes) and had a time frame similar to that measured by Hayward et al. (2004). The post-illumination burst was reproducible (the duration and ion signal intensity were similar) through the daily cycle for the six compounds that exhibited this behaviour.*

L522-524: In the text, the $CO_2$ flux is reported in units of nmol s$^{-1}$ g$^{-1}$, but in Figure 9, the units for the $CO_2$ flux are in nmol h$^{-1}$ g$^{-1}$. One of these units is wrong, please correct it.

*AR: Thanks for catching this. The correct unit is nmol g$^{-1}$ s$^{-1}$. It will be changed in Figure 9.*

L569-571: Furthermore, the value calculated in this work comes from a single Sitka spruce seedling, and therefore may not be an accurate representation.

*AR: We agree that more saplings would be required to have a robust statistical evaluation of interindividual variability. Nevertheless, we still believe the estimation (that BVOCs account for ~0.2 of C assimilation) is useful information and worth reporting. We will reduce section 3.4.1 to a single paragraph integrated into the $CO_2$ flux discussion and place Table 1 in the Supplementary Information.*

L582: Fig. 9 shows the time series of $CO_2$ flux. Please cite the correct figure.

*AR: Thanks for spotting this. We will cite the correct figure in the revised manuscript.*

L582: Do you mean Table S6 and S7?

*AR: Thanks again. Yes, this will now be corrected.*

L599-600: Some monoterpenes originate from de novo synthesis and show both temperature- and light-dependency, e.g., see Ghirardo et al. (2010). You have also shown this in your own results.

*AR: This is true. We will therefore revise the text in this part as follows:*

*"This was unexpected, especially for monoterpenes which are usually assumed to be emitted solely from storage pools (Fuentes et al., 2000) and thus entirely temperature dependent with no PPFD dependence (Hayward et al., 2004). A previous study already observed that monoterpene emissions from several boreal species (Pinus sylvestris, Picea abies, Larix decidua and Betula pendula) may also depend on light due to a significant contribution of de novo emissions (Ghirardo et al., 2010)."*

L686: It might not be appropriate to scale the emission results from a single seedling to determine the BVOC emission fluxes for all Sitka spruce plantations in Ireland. For a more accurate representation, measurements on additional seedlings would be required to account for intraspecific variability.

L705-710: Furthermore, the BVOC emissions of mature trees can differ from those of younger seedlings.

AR: Taking these two comments together. We agree that BVOC emissions from young and mature trees may differ and that scaling the emission results from one tree has limitations. We will therefore revise the discussion section, by significantly reducing the text describing the scaling effort and placing Table 3 in the supplement. But we do believe it is interesting to report that depending on environmental conditions, the dominant BVOC emitted by Sitka spruce could change, and that this effect is due to the non-linear relationship with temperature and light resulting from the specific combination of pooled + de novo pathways. As a result, we plan to retain the discussion about the effect of environmental drivers on the emissions.

 **Supplementary Information:**

Table S2: What do the different colors (red and yellow) represent?

AR: Yellow and red colours help to visually identify compounds emitted by Spruce 1 and 2 respectively. This will be explained in the table caption.

Figure S7 and S8: Please confirm whether the units for the $CO_2$ fluxes are supposed to be in nmol $s^{-1}$ $g^{-1}$ or nmol $h^{-1}$ $g^{-1}$.

AR: The units will be corrected in both figures.

Table S6 and S7: The units for PPFD and emission flux are wrong. Please correct these.

AR: This will be corrected in a revised manuscript.

Technical corrections:
L13: The scientific name (Picea sitchensis) should be italicized.
L25: Replace "Stika spruce" with "Sitka spruce"
L62: Replace "Stika spruce" with "Sitka spruce"
L83: Replace "For identification purposed the trees were named;" with "For identification purposes the trees were named:"
L110: Replace "Viasala" with "Vaisala"
L155: Replace "Agilent 5977B MDS" with "Agilent 5977B MSD"
L297: Replace "Fig 2 and S2" with "Fig 2 and Table S2"
L402: Replace "six BOCs" with "six BVOCs"
L446: Replace "measure data" with "measured data"

L490: Replace "measure data" with "measured data"

Figure 8 caption: Replace "measure data" with "measured data"

L589: Replace "Stika spruce" with "Sitka spruce"

L600: Replace "Stika spruce" with "Sitka spruce"

L729: Replace "Stika spruce" with "Sitka spruce"

L747-751: The units for PPFD are incorrect.

*AR: Thanks very much! All of these technical corrections will be made.*

---

## Author Comment (AC2)

**Reply to Referees' Comments**

**Reviewer 2**

*The authors would like to thank the referee for their detailed review of the manuscript and the helpful comments that will help us to provide an improved version of the manuscript. We address all the comments below and we will further integrate the corresponding changes in a revised article.*

The manuscript quantifies the BVOC emission rates, potentials and blends of Sitka spruce, a species that may be a locally important BVOC source where it is used in plantation forestry. The manuscript highlights the potential of new gas analysers for producing a fuller understanding of total BVOC emissions by including traditionally harder-to-measure oxygenated compounds. With these data, the manuscript aims to produce an estimate for total BVOC emissions from Sitka spruce in Ireland. The question of quantifying BVOC emissions by a tree species that is used in plantation forestry is important, and using two analytical methods for detecting the BVOCs is a valuable contribution. Classifying the compounds by their emission pathway (pool emissions, de novo emissions or combination of both) is also interesting and in line with current research.

The major problem with the methods and interpretation of the results is the small number of study trees. Three to start with is not a lot, and throughout the manuscript, the three trees are reduced to only one. This is a normal problem in biological sciences, but the severe limitations in data should be considered carefully when analysing and using it to form conclusions. Here, the calculations on tree carbon balance and BVOC total emissions are only based on data from one seedling, which is not sufficient to be useful considering that it does not even allow calculating uncertainties in the estimates. Furthermore, the authors themselves also show that there is a large variation in BVOC emission rates between the three trees, so choosing only one of the trees to represent all the trees of the plantations seems weakly justified. Another related problem is the discussion around plant stress. A potential unrecognised stress was used to explain the differences between trees 1 and 2, and based on all the results shown it indeed seems that tree 2 was not performing at the same level as tree 1. Yet, these results are not sufficient to make claims on how stress affects BVOC emissions from Sitka because 1) the stress was unknown and not controlled and 2) again, there are only two trees that are compared to each other, and we cannot know which is "the normal". In other cases, stress can also increase BVOC emissions from pools.

The manuscript is for the most part very easy to follow and contains most of the pertinent information on methods and calculations (for a couple of further questions

on the statistical testing and model selection, see the specific comments below). In particular, the experimental setup was well described which I greatly appreciated. The manuscript results and discussion part is lengthy, especially relative to the small scope of the results. The manuscript could be balanced by condensing and focusing the results and discussion section on one or two questions that can be answered with the given data, and fully removing the parts on carbon balance and BVOC emissions upscaling.

*AR: The authors would like to thank the reviewer for their very useful comments. The aim of the study was to characterise Sitka spruce emissions (identify VOCs and quantify emission rates) and provide a comprehensive analysis of the respective contribution of de novo and pooled emissions. We are conscious that the number of trees would ideally be higher to take into account interindividual variability. Indeed, the intention of our study was to replicate all measurements on three healthy plants. Despite obtaining results from only one healthy tree, we believe the profile of emitted BVOCs and identification of emission pathways are robust as we conducted a comprehensive set of experiments where each cycle (temperature, light and daily cycles) was repeated for at least 7 consecutive days. We accept that using emissions from one healthy tree to provide estimates of emissions for scaling up purposes and carbon balance is quite uncertain. As a result, we will significantly reduce this part of the article in a revised manuscript.*

*Regarding the comments related to stress, we proposed that the difference in species emitted by Spruce 1 and Spruce 2 is best explained by Spruce 2 experiencing some unknown stress. This hypothesis is supported by fluorescence and $CO_2$ measurements. We are pleased that the reviewer agrees that the results do indicate stress in Spruce 2. At the same time, we agree that our results are not sufficient to make claims on how stress affects BVOC emissions from Sitka spruce and will review and re-word the text if necessary to avoid misinterpretation.*

**Specific comments:**

Abstract:

> The abstract is lacking a bit of the motivation for why the study was done, what the overarching objective or question was, or the context of the study

*AR: We will include this information in the abstract of the revised article.*

Introduction:

- I like that the introduction is not too long, but concise and to the point. It also introduces the important concepts for the further manuscript.

*AR: Thank you*

- Line 48: age and stress are not environmental conditions or parameters, rather a status or condition of the plant.

*AR: This will be changed to "environmental factors and plant conditions"*

- Lines 51-60: these could be combined into one paragraph? And the part on standardisation of the BVOC emission rates could be shortened to a simple phrase, e.g. " because of the strong dependence of BVOC emission fluxes on the temperature and PPFD, BVOC emission fluxes are often reported at standard conditions of 30 °C and 1000 µmol m-2 s-1 to facilitate comparison between studies (Guenther et al. 1993)."

*AR: We will reduce this part in the revised manuscript.*

- Lines 39-47, 74: You could make it even more clear to the reader why it is important to know the emission pathway for each compound. I suspect your motivation here is that the emission pathway is important so that the emission fluxes can be better upscaled using T, or T and PAR?

*AR: Yes, absolutely. We will add this detail to the manuscript.*

Materials and methods:

- Line 79-84: So do I understand correctly, these trees were 4-yr old when they were used in the BVOC measurements? Did they have similar or different genetic backgrounds?

*AR: Yes, they were 4-year-old trees, with the same genetic background. This detail will be added to the revised manuscript.*

- Line 90: The PPFD of 250 µmol-2 s-1 feels low. Was this a limitation of the chamber or chosen based on the mean conditions in Ireland?

*AR: This was the maximum PPFD generated in the chamber. However, it also corresponds to the mean light conditions in Ireland.*

- Line 110: "Viasala" -> "Vaisala"

*AR: Thanks. This will be corrected.*

- The growth chamber set-up is very clearly explained, thank you!

*AR: Thank you for the nice comment!*

- Line 121: You could consider moving the section 2.3 Experimental procedures here (after section 2.1), before giving the details on the gas analysers and auxiliary measurements. For me, it would make most sense to first read how the sampling of BVOCs was done, and then read on how they were analysed.

*AR: Following the referee's recommendation, we will move section 2.3 so that it follows on directly from section 2.1, and renumber the sections in Materials and Methods.*

- Line 136: Isoprene was not calibrated as isoprene?

*AR: Unfortunately we could not perform an isoprene calibration experiment. Instead, we extrapolated the average response factor from other hydrocarbons (δ-3-carene and β-myrcene) to isoprene measurements.*

- Line 147: Were there any large peaks (potentially large emissions) that could no be determined, and could thus bias the results? Or were they mostly small peaks?

*AR: The unidentified peaks were mostly small peaks with negative mass defect, meaning they probably originated from low-level contamination as shown in previous studies (Karl et al., 2018; Salazar Gómez et al., 2021, 2019).*

- Line 162: Capital T for "the"

*AR: Thanks, this will be corrected.*

- Line 202: during the adaptation time, was the BVOC enclosure also already installed?

*AR: Yes, the BVOC enclosure was already installed during the acclimatisation period. This will be added to the manuscript.*

- Line 241: The equation is missing the "flow" (through the enclosure)

*AR: Thanks, this will be corrected.*

- Lines 246-249: Did you test this separately for each tree or all trees together? And you used all half-hourly points of tree enclosure and empty enclosure measurements? Was the t-test paired (one tree enclosure measurement corresponded to the empty enclosure measurement closest in time)?

*AR: As one of the original aims was to identify BVOCs emitted for the 3 spruce saplings and compare their emissions (see the individual variability), we tested each tree separately. Then, all half hourly measurements of each tree enclosure were compared to empty*

*enclosure measurements. The measurements for one tree enclosure are supposed to be completely independent other tree enclosures, therefore the t-test was unpaired. Welch's t-test is suited for equal length samples where variance is not presumed to be equal, which was clearly the case here.*

- Line 250: In figure S1 tree 3 does seem quite dead but still it is surprising that you did not get any significant BVOC signal. Do you have a guess why it did not show any significant signal? Did you measure chlorophyll fluorescence and $CO_2$ flux from that tree?

*AR: Yes, it is surprising that Spruce 3 did not emit significant amount of BVOC because we know that even leaf litter from coniferous species has emissions (for example, see Viros et al., 2020). But we do not have an explanation for our observations, other than the concentrations were too small and were below the detection limit of the ToF CIMS. Note that we filtered and analysed only ion signals that are at least 3 times higher in the plant enclosure when compared to the empty enclosure. Some VOCs seemed to be emitted in very low amounts (as detected by ToF-CIMS and TD-GC-MS) but the emissions were not high enough to pass this filter.*

*$CO_2$ fluxes were measured but no uptake or emission was observed (see figure below). Chlorophyll fluorescence was only measured 3 days during the Daily cycle, and Fv/Fm values varied between 0.5 and 0.8.*

[Figure]

*Figure 1: Time series of $CO_2$ flux for Spruce 3 during the Daily Cycle.*

Line 258: For comparing with other emission values in literature, it would be good to also calculate emissions per needle mass or area (as most other studies tend to do). Adding the branch will make a big impact on the calculated emission rate as branch wood is likely much heavier than the needles.

*AR: Emissions are often based on leaf masses because they are expected to only come from leaves. However, it is known for coniferous species which emit BVOCs from storage pools, that the stem and branches can emit significant amount of VOCs, e.g. see Staudt et al.*

*(2019). After all measurements were performed (especially temperature cycles), we found evidence for a strong contribution from the pooled pathway and interpreted this to indicate that the branches could contribute, at least in part, to the measured emissions. This is the reason why we decided to normalise BVOC emissions to the combined mass of the branches and needles. Biomass data are available in Table S1, showing that the difference in biomass is a factor of two. In other words, if our approach was not appropriate, we underestimate emissions by a factor of two.*

*However, for $CO_2$ exchange, we assumed that $CO_2$ assimilation was essentially supported through stomata – that is why $CO_2$ flux was normalised by needle mass only.*

- Line 275: You could give also here the value you used as the maximum enzyme activity temperature

*AR: We used the approach employed by Schuh et al. (1997) for cases where Tm is much higher than the measurement temperature, which is the case here as Tm is generally well above 300 K. in that case, due to the large temperature difference between Tm and the temperatures in this study, the exponential term (equation 5 in the manuscript) can be approximated to zero, and then the denominator can be approximated to one. This was explained in lines 614-620 in the manuscript, which we will move to methods section (see comments below).*

- Line 290: This paragraph is not quite clear to me. By coefficients, do you mean the coefficients for Epooled and Ebiosynthesis as: Ecombined= a*Epooled + b*Ebiosynthesis? Or coefficients within the Epooled and Ebiosynthesis functions? Were the coefficient values you used in this study the exact same ones as Schuh et al 1997, or fitted for your data?

*AR: By coefficients here we meant the empirical coefficients α, cL1, cT2, cT2 and β, which were specific to each spruce sapling.*

Results:

- Line 302 or line 315: Somewhere around here, you could add a sentence on how well the two gas analysers captured the same compounds (shown in Table S2).

*AR: This information will be incorporated into the revised manuscript.*

- Line 308: For which species these examples are? It would be most fruitful to only compare to the same species, or at least the same genus.

*AR: Among the examples cited, Purser et al. (2021) includes Sitka spruce, Janson et al. (1993) Norway spruce (Picea abies), but the Haapanala et al. (2009) publication is on mountain birch (Betula pubescens). We compared our data as much as we can to previous studies on Sitka spruce but the number of studies is limited (only 4 studies so far, Beverland et al.*

*1996, Street et al. 1996, Hayward et al. 2004, Puser et al 2021, all cited in the initial manuscript). We will revise the text accordingly in the revised version.*

- Line 319: There could be a figure or table where the emission rates or contributions of the most common compounds per tree would be shown, also in the main text. Table or a pie chart or bar chart, for example.

*AR: Thanks for this suggestion. We have created a new figure (see below) which will be added to the Supplementary Information.*

[Figure]

*Box plot for Spruce 1 (top) and Spruce 2 (bottom) VOC emissions (in µg $g_{dw}^{-1}$ $h^{-1}$). The bar represents the median, the bottom and top limits of the box are the 25th and 75th quantiles, and the end of the bottom and top bars represent the 5th and 95th quantiles.*

- Line 334: Was the resin running or solid? Was it within the enclosure? Exposed fresh resin can be a huge emission source of monoterpenes and dominate the BVOC emission measurements. If this is the case, the measured values should not necessarily be used for further calculations, because they are strongly biased by the exposed resin. Resin on stem surface is a normal occurrence but quite annoying when trying to measure BVOCs from shoots.

*AR: The resin was more like solid (i.e. not fresh). Although some of the resin spots were in the enclosure, they were small. We believe that they did not contribute a major fraction to emissions as light greatly affected most BVOC emissions, while emissions from resin deposits are only temperature dependent.*

- Lines 347-359: Here and elsewhere in the results/discussion, it would be helpful to the reader if you referred to the tables / figures in the manuscript when describing your results. Here for example, you could refer to Table S2?

*AR: The results and discussion section will be edited to include references to the Tables and Figures where appropriate.*

- Line 362: "six of the detected BVOC.." rather than six of the emissions

*AR: This will be corrected in the revised manuscript.*

- Section 3.2. The figures could be condensed in this section. Figures could cover shorter time spans, for example, 3-4 days - or overlay the days as in Figure 6. The texts also could be condensed a little: avoid describing too many details that are easy to see in the figures but concentrate on the most important general tendencies.

*AR: We think it is important to show that diurnal cycles of BVOC emissions are reproducible on a day-to-day basis, especially because we do not have many saplings. We prefer to leave the figures as they are. The text will be condensed in the revised manuscript.*

- Line 394: could you add a few other compounds in Figure 3 to show the different responses by different compounds

*AR: We will add the traces for another two dominant BVOCs- isoprene and monoterpenes to Figure 3 (see Figure below). Note that another compound is already presented in Figure 4.*

[Figure]

*Time series of a) piperitone b) isoprene and c) monoterpene emission fluxes from Spruce 1 during the Daily Cycle. The colours represent nighttime and daytime, and the intensity of the yellow colour traduces the intensity of light and temperature.*

- Line 402: Figure 4 y-axis says C11H14O

*AR: This will be corrected in the revised manuscript.*

- Line 449: refer to the figure 6

*AR: This will be corrected in the revised manuscript.*

- Line 455: this could be shown in a figure?

*AR: The time series of isoprene emissions during the Temperature cycle will be added in the revised version (see below). The figure clearly shows that when light was turned on and stable, the isoprene signal increased while temperature was maintained at 15˚C.*

[Figure]

*Time series of the isoprene ($C_5H_8$) emission flux from Spruce 2 during the Temperature Cycle.*

- Line 463: You say that the emissions were more strongly influenced by temperature, but is it not rather that they were less influenced by light? I.e., the temperature impact may still be the same.

*AR: This is a good point. The idea was to state that for Spruce 1, PPFD was the main emission driver and for Spruce 2 temperature was the main emission driver. It is true that Spruce 2 BVOC emissions responded less to light, probably because of the mild stress it was suffering (supported by low $CO_2$ assimilation). Then, BVOC emissions were dominated by the pooled emission pathway, and this is why the temperature effect seemed more pronounced for Spruce 2. We will change this sentence in the text accordingly.*

- Line 474: Figure 7 shows that also temperature varied during the Light Cycle. Is this because increasing the light level also increased temperature although you intended the temperature to remain at 18 ºC? With the increase in temperature during the light cycle it is hard to say how much the increases or decreases in emission rate are due to light level changes and how much due to the consequent temperature variations.

*AR: There is a mistake - the legend of Figure 7 was wrong. This will be corrected as shown in the figure below.*

[Figure]

- Line 495: Change the reference format to match the rest of the manuscript

*AR: This will be corrected in the revised manuscript.*

- In order to streamline the manuscript, the CO2 fluxes could be covered in a much shorter format. You could only show the difference in CO2 flux rates between tree 1 and tree 2 to support your hypothesis on the lower level of functioning (stress) of tree 2 (lines 546-553). You could move this part to the beginning of the results section – it would be interesting for the reader to know before looking at the BVOC emission rates that the tree 2 is photosynthesising at a lower level than tree 1. With that, you could also mention that the chlorophyll fluorescence did not differ between the trees, which is surprising. No need to discuss or show the diurnal patterns of CO2 flux, because they are as expected based on respiration and photosynthesis (positive flux in the dark, negative flux in the light).

*AR: This paper (i) focuses on the identification and quantification of key BVOC emissions from Sitka spruce using an unprecedented analytical effort for this species, and ii) provides a comprehensive understanding of BVOC emission pathways (de novo and pooled emissions). We therefore believe it makes sense to first address the question of BVOC emissions in the paper. Then, all other measurements are used to explain the variation in emissions and relate these observations to factors such as photosynthesis, stress, etc. Thus, the authors do not agree with the alternative approach outlined by the referee. However, we accept the point about discussing the $CO_2$ fluxes and diurnal cycles and will reduce the text on this part.*

- Line 521: plant growth produces CO2 because of mitochondrial respiration used as energy source (contributes to positive CO2 flux)

*AR: We think this statement is incorrect. If we define plant growth as an increase in dry mass, there has to be more uptake of $CO_2$ than release in mitochondrial respiration by growing plants which do not get their carbon from roots uptake.*

- Section 3.4.2 This is interesting in the sense that you were able to add many more BVOCs in calculating the carbon balance than normally is possible (with instrumentation that does not capture all compounds you could). However, these results are based on only one tree in lab conditions, measured over a short time period, so the usefulness of the results is very limited. The calculation also includes the bias that some of the BVOC emissions included here do come from pools, so the carbon released in their emissions is carbon that has been captured days or months beforehand.

*AR: Following comments from both referees on this part of the manuscript, we reduced the section on carbon balance to a single paragraph and moved Table 1 to Supplementary Information, as we believe that the values (BVOC emissions represent ca 0.2% of $CO_2$ assimilation flux), at least provide an order of magnitude estimation.*

- Sections 3.5.1: I think these tests for the emission pathway are quite interesting, but the discussion could be shortened. You could consider focusing on what were the proportions of total BVOC emissions that were pool emissions, de novo emissions, or both, and how this differed between your two trees. As you anyway don't show the emission data from the measurement cycles and models for all compounds, you do not need to discuss each of them in a lot of detail. One option could also be to add plots or tables for all compounds in the supporting materials (or actually tables S6 and S7 would already be enough) and guide the reader there in case they are interested in the emission pathway for a specific compound or compound group.

*AR: Thank you. Showing all the plots in the Supplementary Information will make it very hard to read. We propose to include all of them in a separate pdf file, which can be accessed using the online repository cited in the paper (10.5281/zenodo.10514476). In addition, we will shorten the discussion as proposed by the reviewer to simply discuss the fraction of biosynthetic and pooled emission pathways for the compounds.*

- Lines 578-582: This should be added to the methods. In addition, how did you determine which method best reproduced the emission profile? Visually based on the figure or with some goodness-of-fit metrics?

*AR: We will move the mentioned lines to section 2.4 4 Emission calculation and modelling. The chosen model was decided by doing a visual comparison of emission time series for the Daily cycle. We always favoured the simplest solution (single modelled, either pooled or de novo) when it reproduced the measurements. But when it obviously did not match, the combined emission model was used. For example, it is clear that all BVOCs with night time emissions cannot be reproduced by a de novo emission model, and BVOCs experiencing an increase during the Light cycle cannot be reproduced by the pooled emissions model.*

- Line 615-620: This should be in the methods

*AR: We will move these lines to methods.*

- Line 629: Did you calculate a correlation or is this based on the visual assessment of the figure?

*AR: As explained above, this is based on the visual assessment of time series.*

- Line 635: for comparisons, you could pull out other studies on conifers (with monoterpene pools in needles), for example see Ghirardo et al. 2010 (https://doi.org/10.1111/j.1365-3040.2009.02104.x)

*AR: This particular reference will be used in the text and cited.*

Section 3.5.2: I understand the wish to try and upscale the BVOC emissions to see potential total emissions from the Sitka spruce plantations. However, with the emission data that is only based on one seedling, you cannot even really get an estimate for the uncertainty in the calculation. I would propose adding the comparisons from Table 2 at the end of the previous section and removing the emission upscaling part of the manuscript.

*AR: As explained above, we agree that the section on the upscaling and C balance should be reduced significantly. However, we believe that is useful to report the order of magnitude for the emissions. We will therefore considerably reduce this section to include a simple discussion only.*

Figures:

- Figure 1 is really nice, clear and helpful for understanding the measurement system!

*AR: Thank you.*

- Figure 2: is this showing results from Tof-CIMS and TD-GS-MS or one of them? Clarify that in the legend. In this figure, the downward columns make me

think of negative emissions (deposition), although of course that is not what the figure wants to show. Consider dividing the figure into two subfigures, one per tree, that are stacked one on top of the other and that have the y-axis going from 0 to 17 from bottom up. So, flipping tree 2 around. This would help to avoid misreading the figure, which otherwise is nice and a good idea on how to show the emission spectra.

*AR: The results in Figure 2 are from both TD-GC-MS and ToF-CIMS. The reviewer is correct in that the negative part of the figure can be interpreted as deposited BVOCs, we will therefore follow the suggestion and split the figure. The Figure caption will also be edited accordingly.*

- Figures 3-5, 9: these don't need to show the whole time series, and I'd recommend also doing the same as you did in Figures 6 and 7 – overlay all days in one figure

*AR: One of the strengths of our study is that we follow emissions over 7 consecutive days. As a result, we believe it is important to present this data as full time series to show the day-to-day variation and reproducibility as appropriate.*

- Tables S6 and S7: it would be interesting if these two tables were combined, it would allow better comparison between the compounds

*AR: These Tables will be combined in a revised version.*

References

Byrne, K.A., Kiely, G., Leahy, P., 2005. CO2 fluxes in adjacent new and permanent temperate grasslands. Agric. For. Meteorol. 135, 82–92. https://doi.org/10.1016/j.agrformet.2005.10.005

Ghirardo, A., Koch, K., Taipale, R., Zimmer, I., Schnitzler, J.-P., Rinne, J., 2010. Determination of de novo and pool emissions of terpenes from four common boreal/alpine trees by 13CO2 labelling and PTR-MS analysis. Plant Cell Environ. 33, 781–792. https://doi.org/10.1111/j.1365-3040.2009.02104.x

Hayward, S., Tani, A., Owen, S.M., Hewitt, C.N., 2004. Online analysis of volatile organic compound emissions from Sitka spruce (Picea sitchensis). Tree Physiol. 24, 721–728. https://doi.org/10.1093/treephys/24.7.721

Karl, T., Striednig, M., Graus, M., Hammerle, A., Wohlfahrt, G., 2018. Urban flux measurements reveal a large pool of oxygenated volatile organic compound emissions. Proc. Natl. Acad. Sci. 201714715. https://doi.org/10.1073/pnas.1714715115

Lavi, A., Vermeuel, M.P., Novak, G.A., Bertram, T.H., 2018. The sensitivity of benzene cluster cation chemical ionization mass spectrometry to select biogenic terpenes. Atmospheric Meas. Tech. 11, 3251–3262. https://doi.org/10.5194/amt-11-3251-2018

Portillo-Estrada, M., Zenone, T., Arriga, N., Ceulemans, R., 2018. Contribution of volatile organic compound fluxes to the ecosystem carbon budget of a poplar short-rotation plantation. GCB Bioenergy 10, 405–414. https://doi.org/10.1111/gcbb.12506

Salazar Gómez, J.I., Klucken, C., Sojka, M., Masliuk, L., Lunkenbein, T., Schlögl, R., Ruland, H., 2019. Elucidation of artefacts in proton transfer reaction time-of-flight mass spectrometers. J. Mass Spectrom. 54, 987–1002. https://doi.org/10.1002/jms.4479

Salazar Gómez, J.I., Sojka, M., Klucken, C., Schlögl, R., Ruland, H., 2021. Determination of trace compounds and artifacts in nitrogen background measurements by proton transfer reaction time-of-flight mass spectrometry under dry and humid conditions. J. Mass Spectrom. 56. https://doi.org/10.1002/jms.4777

Schuh, G., Heiden, A.C., Hoffmann, Th., Kahl, J., Rockel, P., Rudolph, J., Wildt, J., 1997. Emissions of Volatile Organic Compounds from Sunflower and Beech: Dependence on Temperature and Light Intensity. J. Atmospheric Chem. 27, 291–318. https://doi.org/10.1023/A:1005850710257

Staudt, M., Byron, J., Piquemal, K., Williams, J., 2019. Compartment specific chiral pinene emissions identified in a Maritime pine forest. Sci. Total Environ. 654, 1158–1166. https://doi.org/10.1016/j.scitotenv.2018.11.146

Viros, J., Hernandez, C., Wortham, H., Gavinet, J., Lecareux, C., Ormeño, E., 2020. Litter of mediterranean species as a source of volatile organic compounds. Atmos. Environ. 242, 117815. https://doi.org/10.1016/j.atmosenv.2020.117815